# Proton-assisted creation of controllable volumetric oxygen vacancies in ultrathin $CeO_{2-x}$ for pseudocapacitive energy storage applications

Sajjad S. Mofarah [1], Esmaeil Adabifiroozjaei [2], Yin Yao[3], Pramod Koshy[1], Sean Lim[3], Richard Webster [3], Xinhong Liu [1], Rasoul Khayyam Nekouei[1], Claudio Cazorla [1], Zhao Liu[1], Yu Wang [4], Nicholas Lambropoulos[1] & Charles C. Sorrell[1]

Two-dimensional metal oxide pseudocapacitors are promising candidates for size-sensitive applications. However, they exhibit limited energy densities and inferior power densities. Here, we present an electrodeposition technique by which ultrathin $CeO_{2-x}$ films with controllable volumetric oxygen vacancy concentrations can be produced. This technique offers a layer-by-layer fabrication route for ultrathin $CeO_{2-x}$ films that render $Ce^{3+}$ concentrations as high as ~60 at% and a volumetric capacitance of 1873 F $cm^{-3}$, which is among the highest reported to the best of our knowledge. This exceptional behaviour originates from both volumetric oxygen vacancies, which enhance electron conduction, and intercrystallite water, which promotes proton conduction. Consequently, simultaneous charging on the surface and in the bulk occur, leading to the observation of redox pseudocapacitive behaviour in $CeO_{2-x}$. Thermodynamic investigations reveal that the energy required for oxygen vacancy formation can be reduced significantly by proton-assisted reactions. This cyclic deposition technique represents an efficient method to fabricate metal oxides of precisely controlled defect concentrations and thicknesses.

[1] School of Materials Science and Engineering, UNSW Sydney, Sydney, NSW 2052, Australia. [2] Research Center for Functional Materials (RCFM), National Institute for Materials Science (NIMS), Tsukuba, Ibaraki 305-0047, Japan. [3] Electron Microscopy Unit (EMU), Mark Wainwright Analytical Centre, UNSW Sydney, Sydney, NSW 2052, Australia. [4] Mark Wainwright Analytical Centre, UNSW Sydney, Sydney, NSW 2052, Australia. Correspondence and requests for materials should be addressed to S.S.M. (email: s.seifimofarah@unsw.edu.au) or to E.A. (email: adabifiroozjaei.e@nims.go.jp) or to P.K. (email: koshy@unsw.edu.au)

Increasing demands for energy-efficient portable electronic devices of minimal size have led to the development of two-dimensional (2D) electrochemical pseudocapacitors (EPCs)[1–4]. Fabrication of 2D EPC materials of adequate transparency and electrical conductivity can extend the potential applications to optoelectronic devices, such as touch screens and solar cells[5,6]. Generally, the two-dimensionality of the structures provides ultrafast electron transferability across the films and high-charge carrier accessibility to the active sites, resulting in desirable energy and power densities. Among the different types of energy storage devices, metal oxide (MO) pseudocapacitors have shown superiority owing to their greater energy densities and higher power densities compared to those of electrical double-layer capacitors (EDLCs) and batteries, respectively[7]. However, conventional synthesis methods, which involve assembly of 2D MO nanosheets, lead to tight stacking of the sheets in the film, thereby hindering charge carrier diffusion through the interlayer spaces, impacting negatively on performance[8–10]. As an alternative, MOs with intercalation pseudocapacitive behaviour (e.g., $RuO_2$ and $Nb_2O_5$) can be used since charge/discharge occurs not only at the electrode/electrolyte interface but also within the bulk of the 2D films[11–13]. Nonetheless, high costs and toxicity in the case of $RuO_2$ and poor electron transfer rates in the case of $Nb_2O_5$ limit their applicability[14–16]. Further, limitations in achieving reproducible film thicknesses have resulted in inconsistent performance. To the best of our knowledge, intercalation pseudocapacitance has not been observed to date in $CeO_2$, although very high specific capacitances of this material have been reported previously[17,18], indicating its promise for energy storage applications.

Here, we show a novel cyclic electrodeposition method to synthesise ultrathin films of $CeO_{2-x}$ (9–70 nm) comprised of ultrafine crystallites (3–8 nm). These films exhibit high oxygen vacancy concentrations ($[V_O^{\bullet\bullet}]$) of ~4–15 at% and intercrystallite $H_2O$, where the former enhance the electron conduction and the latter promotes the proton conduction. These $CeO_{2-x}$ ultrathin films demonstrate proton insertion/disinsertion pseudocapacitive behaviour. They exhibit the outstanding volumetric capacitance of 1873 F cm$^{-3}$, which considerably exceeds the highest value yet reported (1160 F cm$^{-3}$ for ultrathin $MnO_2$/Au)[19] to the best of our knowledge. This work reports a simplified, scalable and controllable method that can be extended to the fabrication of other MO thin films that are being used increasingly for electronic, energy storage and photoelectrochemical applications.

## Results

**Characterisation of thin films**. The thermodynamics of the $Ce^{3+}$-$Ce^{4+}$-$CH_3COOH$-$H_2O$ system, which was used to fabricate the thin films, were investigated (details in Supplementary Note 1). The resultant data were analysed to understand the electrochemical behaviour of the system and subsequently to determine the optimal conditions for the electrodeposition of Ce(OH)$_4$, which transforms readily to $CeO_2$ in aqueous solutions[20] (details in Supplementary Note 2). By varying the cyclic voltammetry scan rate and number of cycles, $CeO_{2-x}$ films of different thicknesses and $Ce^{3+}$ concentrations ($[Ce^{3+}]$) of ~18–60 at% (equivalent to 4.5–15% of $[V_O^{\bullet\bullet}]$) were deposited on fluorine-doped tin oxide (FTO) glass substrates. Figure 1a shows a representative X-ray diffraction (XRD) pattern of a $CeO_{2-x}$ thin film. The nanostructure and elemental composition of the film was studied using scanning electron microscopy (SEM), transmission electron microscopy (TEM) and energy dispersive spectroscopy (EDS) (Supplementary Note 3). Figure 1b shows an X-ray photoelectron spectroscopy (XPS) spectrum for the 3d orbital of Ce. The $[Ce^{3+}]$, which was calculated from the areas of

the $Ce^{3+}$ doublet peaks (purple) at the binding energies of ~800 and ~805 eV, was high at 43.6 at% ($[V_O^{\bullet\bullet}] = 10.9\%$). Details of the quantitative analyses of the spectra are provided in Supplementary Note 4. These calculations were correlated with the XPS data for the 1s orbital of the O peaks, as shown in Fig. 1c. These were fit using Gaussian functions at 529.21 eV (peak 1, blue), 531.23 eV (peak 2, purple) and 532.96 eV (peak 3, black). Peaks 1 and 2 are for oxygen bound to $Ce^{4+}$ ($[O]_{Ce^{4+}}$) and $Ce^{3+}$ ($[O]_{Ce^{3+}}$), respectively, and peak 3 is for adsorbed water molecules ($[O]_{H_2O}$)[21]. The area of the peak for oxygen bound to $Ce^{3+}$ is 56.0 at% of the total oxygen concentration ($[O]$), which is even greater than the equivalent value of 43.6 at% for $[Ce^{3+}]$. The difference in values is attributed to the effect of the sensitivity of the XPS beam: For the Ce 3d peak, the penetration depth extends to ~1.0 nm but, for the O 1s peak, it extends to ~1.5 nm[22]. Figure 1d shows a representative high-angle annular dark-field (HAADF) image, which reveals a film thickness of ~18 nm and crystallites in the size range 3–8 nm. Two $[O]$ distributions across (Fig. 1e) and within (Fig. 1f) crystallites were observed by EDS line profiles. Sharp decreases at the crystallite interfaces (Fig. 1e) confirmed increasing $[V_O^{\bullet\bullet}]$ at these locations, as reported previously[23]. In contrast, the EDS line profile of $[O]$ along a single crystallite (Fig. 1f) showed much less variation. The formation of a highly oxygen-deficient $CeO_{2-x}$ structure is confirmed by analysing the relative change in the $Ce^{3+}/Ce^{4+}$ ratio using electron energy loss spectroscopy (EELS). Two representative areas within the bulk were analysed (Supplementary Fig. 7), the spectra of which are shown in Fig. 1g, at the crystallite boundaries, and Fig. 1h, within the crystallite. Hojo et al.[23] showed that the $[V_O^{\bullet\bullet}]$ can be evaluated from the ratio of the intensity (height) of the M5 peak (orange) to that of the M4 peak (green), where the ratio is bounded by the minimal value of ~0.9 for stoichiometric $CeO_2$ (viz., $[V_O^{\bullet\bullet}] = 0\%$) and the maximal value of ~1.25 for $Ce_2O_3$ (viz., $CeO_{2-x}$ with the theoretically maximal $[V_O^{\bullet\bullet}] = 25\%$). In this work, the M5/M4 ratios for boxes g and h were determined to be ~1.19 and ~1.10 and hence represent approximate $[V_O^{\bullet\bullet}]$ of 20.7% and 14.2%, respectively. The greater $[V_O^{\bullet\bullet}]$ at the crystallite boundary compared to that in the bulk is consistent with the results of others[23–27]. However, the high $[V_O^{\bullet\bullet}]$ at intracrystallite regions and the sub-nanometre layer-by-layer deposition (discussed subsequently) reveal that $V_O^{\bullet\bullet}$ are at the boundaries but also suggest a homogenous distribution through the film. These data demonstrate that relatively high $[V_O^{\bullet\bullet}]$ are accommodated at both intercrystallite and intracrystallite regions.

**Mechanism of thin-film deposition**. In order to clarify the mechanism of formation of such ultrathin $CeO_{2-x}$ films, electrodeposition data were deconvoluted in terms of the reactions over cycling (the pH effect analysis is in Supplementary Note 5). Figure 2 shows the voltammograms obtained for cycle numbers 1, 25 and 50 at a scan rate of 300 mV s$^{-1}$. As shown in Fig. 2a, during the forward scan of the 1st cycle, Ce(OH)$_4$ precipitation occurs at $E = 0.47$ V vs. Ag/AgCl (Ox$_1$). This is followed by partial transformation of Ce(OH)$_4$ to $CeO_2$, which is a non-faradaic reaction and thus has no corresponding peak. The cathodic reaction for Ce(OH)$_4$ involves reductive dissolution to Ce(III) at $E = -0.02$ V vs. Ag/AgCl (Re$_1$) and its return to the electrolyte. The lower peak current density of Re$_1$ relative to that of Ox$_1$ indicates only partial reduction of Ce(OH)$_4$, since some of this already is transformed to $CeO_2$. The cathodic reaction for $CeO_2$ involves the generation of $Ce^{3+}$ and $V_O^{\bullet\bullet}$ at $E = -0.13$ V vs. Ag/AgCl (Re$_2$). As Fig. 2b (25th cycle) shows, the as-formed $V_O^{\bullet\bullet}$ are annihilated partially during the forward scan at $E = +0.07$ V vs. Ag/AgCl (Ox$_2$), so there remain some residual $[V_O^{\bullet\bullet}]$ in the film (discussed in next paragraph). This is confirmed

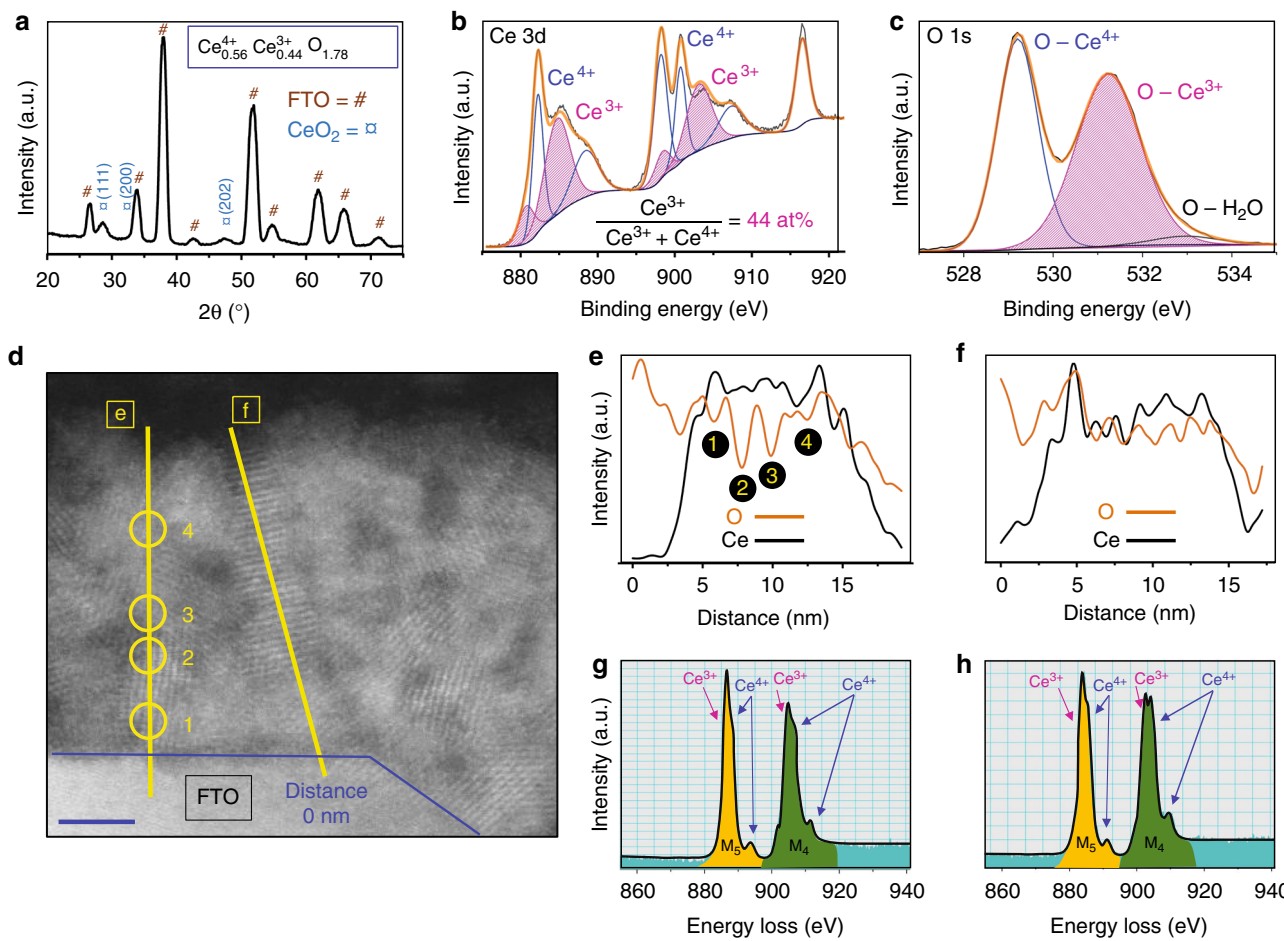

**Fig. 1** Characterisation of CeO$_{2-x}$ thin films. **a** Representative XRD pattern of a CeO$_{2-x}$ film deposited at a scan rate of 3000 mV s$^{-1}$ for 1000 cycles. **b** XPS spectrum for Ce 3d orbital. **c** XPS spectrum for O 1s orbital. **d** HAADF image of nanostructure (scale bar is 2 nm). **e** EDS line scan across crystallite boundaries (left). **f** EDS line scan within crystallite (right). **g** EELS spectrum at crystallite boundary (point g). **h** EELS spectrum within crystallite (point h)

quantitatively by XPS measurements for the films fabricated at the termination potentials at −0.6 V (end of cathodic reactions) and +0.8 V (end of anodic reactions), as shown in Supplementary Fig. 9a, b and c, d, respectively. Also, the increasing growth of the voltammogram area with cycling is indicative of continuous growth of the film[28,29]. Interestingly, further formation of $V_O^{\bullet\bullet}$ results in the emergence of another pair of peak potentials (blue) in Fig. 2b at $E = -0.3$ V vs. Ag/AgCl (Ox$_3$) and $E = -0.4$ V vs. Ag/AgCl (Re$_3$), which correspond to proton insertion in and exclusion from, respectively, the $V_O^{\bullet\bullet}$-rich nanostructure[13,30,31]. These data suggest that $V_O^{\bullet\bullet}$ plays a critical role as a trapping site for proton diffusion during charging[32,33], so increasing the volumetric $[V_O^{\bullet\bullet}]$ should enhance proton insertion/disinsertion and consequently increase the proton-related peak intensities. Thus, expansion in the sum of the peak areas shown in Fig. 2c (at the 50th cycle) can be ascribed to both the increasing thickness and the volumetric $[V_O^{\bullet\bullet}]$ of the thin films. Simulated nanostructures corresponding to Ce(OH)$_4$ formation, Ce(OH)$_4$ to CeO$_2$ transformation and CeO$_{2-x}$ formation are shown in Fig. 2d–f, respectively. Also, comprehensive discussion regarding the peak identification, chemical reactions involved and calculation of the peak areas is provided in Supplementary Note 6.

The formation of $V_O^{\bullet\bullet}$ was confirmed further by thermodynamic calculations (Supplementary Note 7). Typically, the $V_O^{\bullet\bullet}$ formation energy is highly positive and can be varied in the range 1.20–2.25 eV[34], which is subject to the crystal size and exposed facets. However, integration of protons (H$^+$) during reaction in

an aqueous solution can shift dramatically the required energy toward low values of ~0.1 eV (Supplementary Table 9). In our experiments, local protons are provided at the working electrode/electrolyte interface upon OH$^-$ consumption during the formation of Ce(OH)$_4$ (Ox$_1$). Although the durability of as-synthesised $V_O^{\bullet\bullet}$ is affected by the Ox$_2$ reaction, during the forward scan, high OH$^-$ consumption by the Ox$_1$ reaction limits the availability of OH$^-$ for the Ox$_2$ reaction. Thus, the annihilation of $V_O^{\bullet\bullet}$ occurs partially, leaving the remaining $V_O^{\bullet\bullet}$ in the structure. This is confirmed by increasing the peak current density for the proton intercalation reaction (Re$_3$) as a function of cycling. Therefore, both experimental and theoretical evidence confirm a proton-assisted oxygen vacancy creation (PAOVC) mechanism, in which a high content of volumetric $V_O^{\bullet\bullet}$ in the CeO$_{2-x}$ nanostructure can be obtained. There appears to be only one work[35] on the creation of $V_O^{\bullet\bullet}$ using PAOVC. In that work, a constant reduction potential was applied to as-synthesised nanotubes of TiO$_2$ (~40 nm wall thickness), leading to the creation of a high $[V_O^{\bullet\bullet}]$. Although that work followed the same principle as ours, applying an excessively high constant potential to as-synthesised samples results in a situation of highly unstable $V_O^{\bullet\bullet}$, which are in a non-equilibrium condition. Therefore, the $[V_O^{\bullet\bullet}]$ reduces very quickly (over a few hours) in ambient conditions, making such nanostructures unsuitable for applications where long-term exposure is required. In contrast, in our work, there is only a one-step process comprised of cyclic deposition, followed by PAOVC, to yield a high density of volumetric $[V_O^{\bullet\bullet}]$. Additionally,

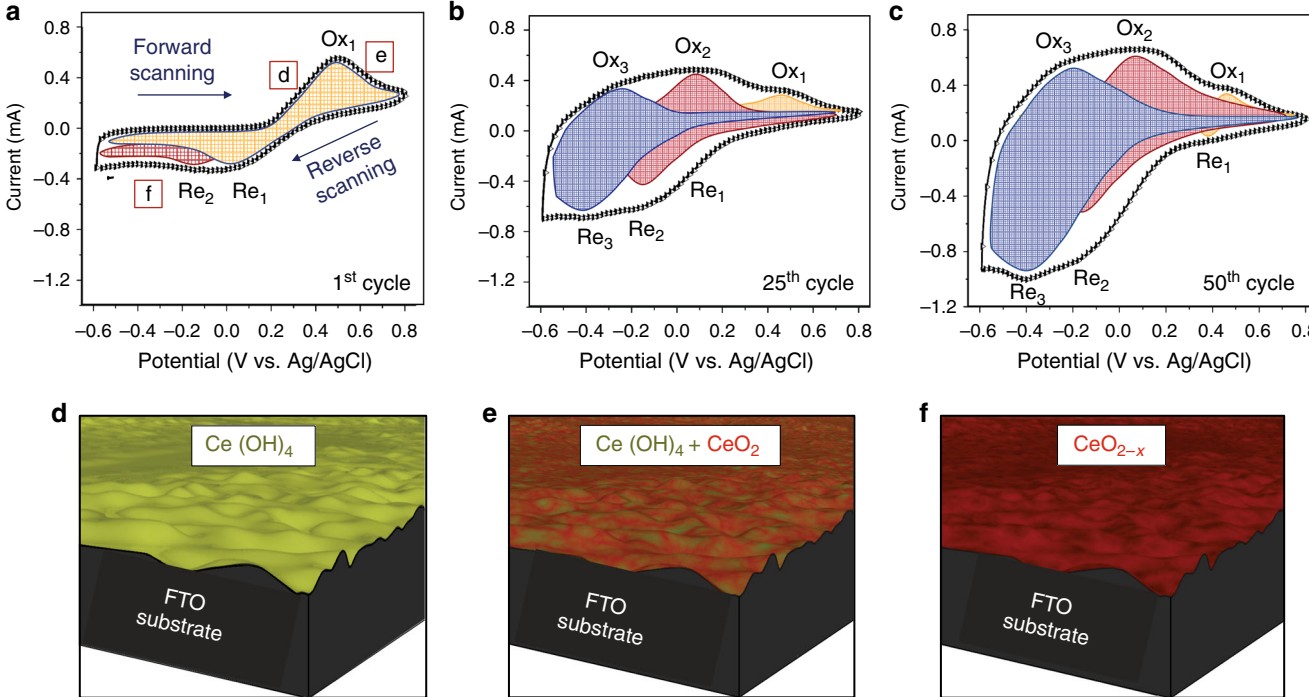

**Fig. 2** Mechanism of CeO$_{2-x}$ thin-film deposition. Voltammograms obtained at a scan rate of 300 mV s$^{-1}$: **a** 1st cycle. **b** 25th cycle. **c** 50th cycle. **d** Schematic of Ce(OH)$_4$ nanostructure deposited during forward scanning. **e** Schematic of mixed Ce(OH)$_4$ + CeO$_2$ nanostructure during chemical (non-faradaic) conversion during forward scanning. **f** Schematic of CeO$_{2-x}$ nanostructure formed during reverse scanning

due to equilibration (production/annihilation) of $V_O^{\bullet\bullet}$ over the cycling, the resultant CeO$_2$ exhibits great chemical stability after long-term exposure to aqueous solutions (Supplementary Fig. 10).

**Electrochemical performances of thin films**. The electrochemical behaviour of the CeO$_{2-x}$ films was investigated in 1 M NaCl aqueous electrolyte (pH = 7) using a three-electrode configuration system, in which Ag/AgCl, Pt coil and deposited film on FTO were used as reference, counter and working electrodes, respectively. Figure 3a shows typical voltammograms of the CeO$_{2-x}$ films with ~44 at% Ce$^{3+}$ (scan rate: 5–500 mV s$^{-1}$). At a scan rate of 5 mV s$^{-1}$, an exceptional volumetric capacitance of 1873 F cm$^{-3}$ and an areal capacitance of 4.56 mF cm$^{-2}$ were achieved (the calculation method is given in Supplementary Note 8). The high capacitances of the thin films are attributed to the optimal crystallite sizes (3–8 nm; Supplementary Fig. 11), which are identical to the experimental particle size range reported (3–8 nm)[36] to exhibit the greatest surface Ce/O ratio and consequent high oxygen storage capacity. In contrast, a subnanometre particle size has been projected by calculation to exhibit optimal oxygen storage capacity[37]. The rapid kinetics of the charge/discharge reactions were assessed by measuring the overpotential values, which did not vary significantly in the range 0.05–0.3 V for all of the scan rates. The rate-controlling mechanism during the charge/discharge is determined by Eq. 1[38]:

$$i = av^b \tag{1}$$

where $i$ is the peak current, $a$ is a constant, $v$ is the scan rate and the exponent $b$ indicates the predominant kinetics mechanism. A $b$ value of 0.5 indicates a slow semi-infinite current while a value of 1 indicates rapid surface-confined capacitive behaviour (i.e., high power density). The $b$ values were calculated to be 0.86 and 0.80 for the charge and discharge reactions, respectively, at scan rates of 5–500 mV s$^{-1}$, demonstrating that the high power density

of the CeO$_{2-x}$ films originated from unique surface-controlled kinetics. Examination of the $b$ values for the well-known intercalation pseudocapacitor Nb$_2$O$_5$[38] (0.80 and 0.70 for the charge and discharge reactions, respectively, at scan rates ≤50 mV s$^{-1}$) indicates that the power densities are comparable.

Additionally, the contribution of different charge/discharge mechanisms (surface: $k_1v$; bulk diffusion: $k_2v^{0.5}$) of the CeO$_{2-x}$ film is considered using Eq. 2[39]:

$$i(V) = k_1v + k_2v^{0.5} \tag{2}$$

where $i(V)$ is the instantaneous current and $k_1$ and $k_2$ are obtained from the slope and ordinate intercept ($v \to \infty$), respectively, of the $i(V)/v^{0.5}$ vs. $v^{0.5}$ plot. Figure 3c shows that the relative contributions of the two mechanisms change at different scan rates, although the surface capacitive mechanism is dominant at all scan rates (Fig. 3c).

Figure 3d illustrates the excellent stability of the charge-storage performance, where the capacitance retention of ~94% after 1000 cycles is achieved. As expected, the efficiency of the film changes slightly during the first 500 cycles and then the efficiency becomes constant. In order to evaluate the stability of the film, XPS analysis was conducted after 1000 cycles, the results of which reveal that there is insignificant change in the [Ce$^{3+}$] while the [O]$_{Ce^{3+}}$ decreased and the [O]$_{H_2O}$ increased, as shown in Supplementary Fig. 10. Figure 3e compares the volumetric and areal capacitances of the CeO$_{2-x}$ film, as a function of thickness, in comparison to other films reported in recent publications. These data reveal that the CeO$_{2-x}$ film exhibits the highest volumetric capacitance and that the value is eight times greater than that of carbon nanotube-MnO$_2$ hybrid ultrathin films[26]. This extraordinary performance can be attributed to the coexistence of (1) effective intracrystallite electron conduction owing to the $V_O^{\bullet\bullet}$ in CeO$_{2-x}$ and (2) rapid intercrystallite proton transfer activated by the adsorbed water molecules (H$_2$O peak in Fig. 1c). The latter is supported by the symmetry of the

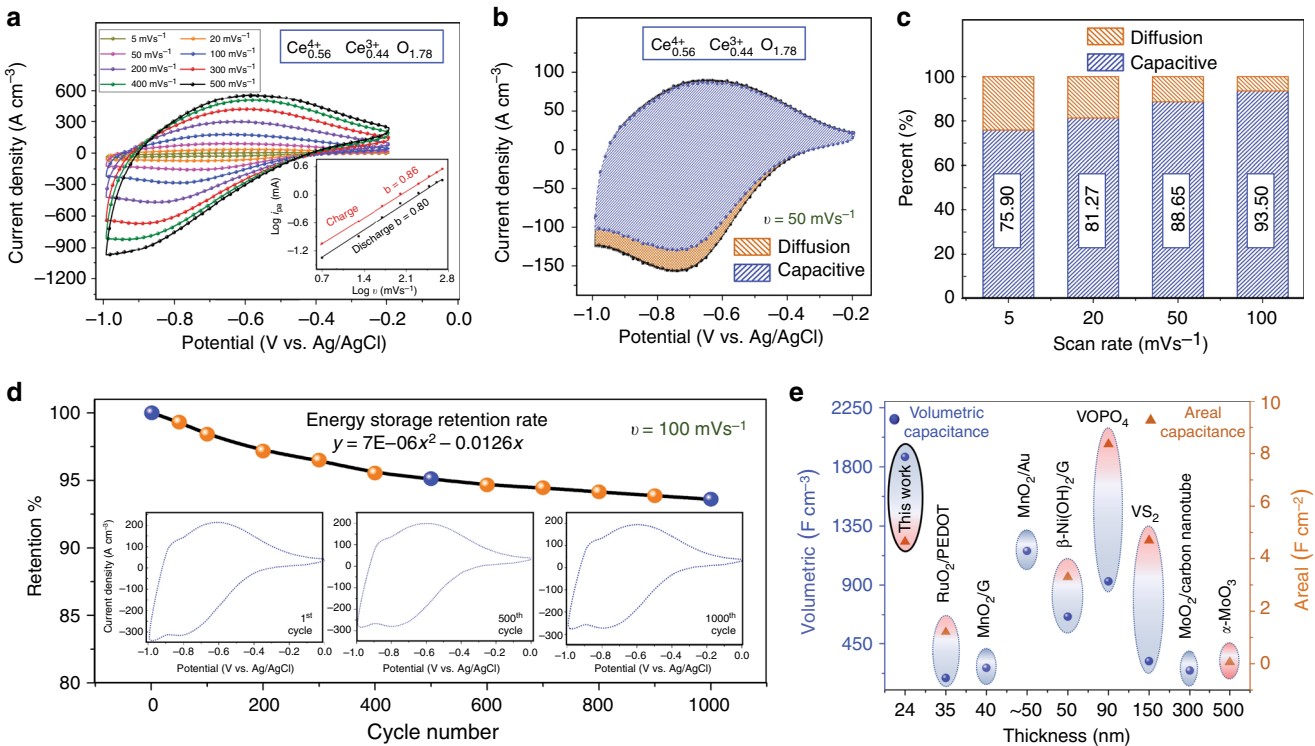

Fig. 3 Pseudocapacitance performance of CeO$_{2-x}$ thin films. **a** Cyclic voltammograms obtained at scan rates in the range 5–500 mV s$^{-1}$, plot of log $i$ vs. log $\upsilon$ in inset. **b** Capacitive and diffusional contributions to the total capacitance obtained at a scan rate of 50 mV s$^{-1}$. **c** fractions of capacitive and diffusional contributions at scan rates in the range 5–100 mV s$^{-1}$. **d** Stability of charge-storage performance up to 1000 cycles at a scan rate of 100 mV s$^{-1}$ (inset shows voltammograms obtained at cycle numbers 1, 500 and 1000). **e** Plot of thickness vs. volumetric and area capacitances obtained in the present work (5 mV s$^{-1}$) compared to data from recently reported works for high-performance materials: ultrathin films of hybrid RuO$_2$/PEDOT[55], MnO$_2$/graphene hybrid nanostructure[56], MnO$_2$/Au hybrid nanostructure[19], ß-Ni(OH)$_2$/graphene hybrid nanostructure[57], VOPO$_4$ thin film[58], VS$_2$ thin film[59], MnO$_2$/multiwall carbon nanotube hybrid structure[28], α-MoO$_3$[14]

voltammograms, particularly the peak potentials of the redox reactions, revealing unique redox pseudocapacitive behaviour that is characteristic of redox pseudocapacitors, such as hydrous RuO$_2$[29,40]. The most commonly cited possible charge/discharge redox reactions are as follows[41]:

$$\text{Cathodic charge reaction} \quad \text{Ce}^{4+}_{1-x}\text{Ce}^{3+}_x\text{O}_{2-y} + \alpha e^- + \beta\text{H}^+ \\ \leftrightarrow \text{Ce}^{4+}_{1-(x+\alpha)}\text{Ce}^{3+}_{x+\alpha}\text{O}_{2-(y+\delta)}\text{OH}_{(\beta+y)} \quad (3)$$

$$\text{Anodic discharge reaction} \quad \text{Ce}^{4+}_{1-(x+\alpha)}\text{Ce}^{3+}_{x+\alpha}\text{O}_{2-(y+\delta)}\text{OH}_{(\beta+y)} \\ \leftrightarrow \text{Ce}^{4+}_{1-x}\text{Ce}^{3+}_x\text{O}_{2-y} + \alpha e^- + \beta\text{H}^+ \quad (4)$$

**First-principles calculations of effect of oxygen vacancies on thin films.** The mechanism of electron conduction has been investigated with spin-polarised density functional theory (DFT) calculations (for details, see Methods). In accordance with recent work[42,43] and DFT calculations[44], the Ce electrons in orbitals 5$s$, 5$p$, 6$s$, 5$d$ and 4$f$ and the O electrons in orbitals 2$s$ and 2$p$ were considered to be the valence electrons. First, we determined the most energetically favourable configurations for $V_O^{\bullet\bullet}$. We found that the interactions between oxygen vacancies are attractive and so they tend to form clusters. According to our DFT calculations, during the transformation from stoichiometric CeO$_2$ (Fig. 4a) to non-stoichiometric CeO$_{2-x}$ (Fig. 4b), the vacancy clusters form one-dimensional tunnel-like structures along the [001] direction. As shown in Fig. 4c, an increase in $[V_O^{\bullet\bullet}]$ from 0% to 25% results in a substantial decrease in the energy band gap ($E_g$) of 38%,

namely, from 3.2 eV[45] to 1.9 eV. Consequently, an increase in $[V_O^{\bullet\bullet}]$ may yield significant enhancements in the electronic transport of CeO$_{2-x}$, as has been suggested previously by other researchers[23,45,46]. The origin of the decrease in the band gap stems from the two electrons left by the $V_O^{\bullet\bullet}$, which fill the localised 4$f$ orbitals in the two neighbouring Ce ions (mostly forming the bottom of the conduction band)[47]. Figure 4d illustrates the electronic charge density redistribution that occurs during the change from stoichiometric CeO$_2$ to non-stoichiometric CeO$_{1.5}$, which leads to a charge density increase around the Ce ions (see surface denoted by blue colour). In addition, we checked how possible nano-size effects influence the estimation of the $E_g$ in stoichiometric and non-stoichiometric ceria and the results, which are given in Supplementary Note 9, reveal insignificant nano-size effects.

**Controllability of thickness and oxygen vacancy concentration of thin films.** In this section, we show that our method can be used for precision engineering of both the $[V_O^{\bullet\bullet}]$ and the film thicknesses. Figure 5a–d show the voltammograms of four films at scan rates of 50, 300, 1000 and 3000 mV s$^{-1}$, respectively. The characterisation data (XRD, laser Raman microspectroscopy (Raman), TEM, atomic force microscopy (AFM) and time-of-flight secondary ion mass spectrometry (TOFSIMS)) of the films are provided in Supplementary Note 10. As expected, increasing the scan rate results in increasing peak area owing to the greater current density and resultant deposition of CeO$_2$. However, owing to the faster kinetics at higher scan rates, there is insufficient time for the Ce(OH)$_4 \rightarrow$ CeO$_2$ transformation to occur immediately before the reverse scan (Fig. 2), which results in the

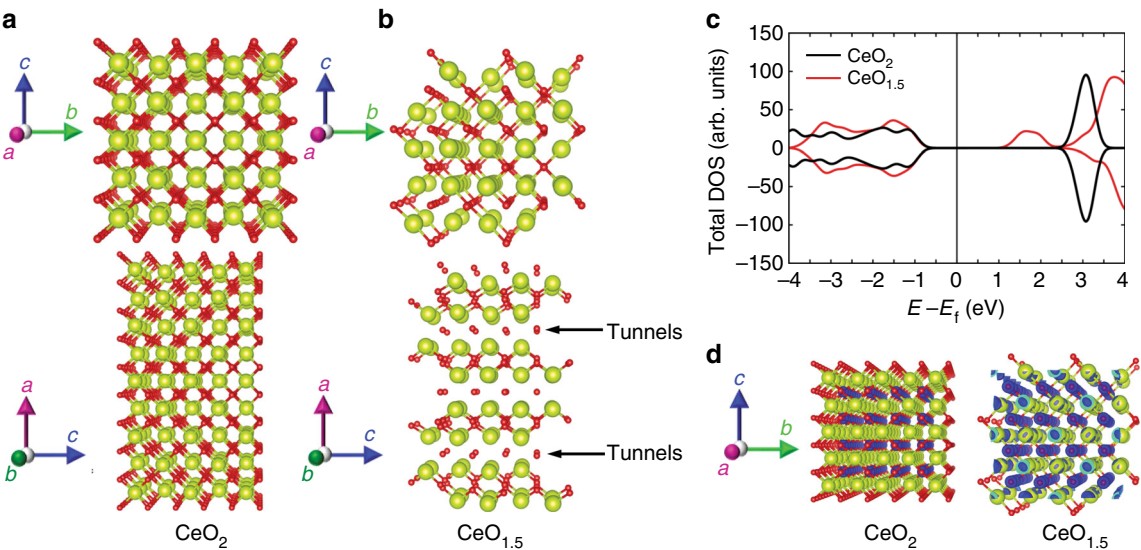

**Fig. 4** First-principles calculations performed on stoichiometric $CeO_2$ and non-stoichiometric $CeO_{1.5}$. **a** Structural projections along the [100] and [110] directions for stoichiometric $CeO_2$. **b** Structural projections along the [100] and [110] directions for non-stoichiometric $CeO_{1.5}$, where the $V_O^{\bullet\bullet}$ are self-organised into clustered one-dimensional tunnels oriented in the [001] direction. **c** Effect of $[V_O^{\bullet\bullet}]$ on calculated spin-polarised total density of electronic states, showing spin-up (positive) and spin-down (negative) regions (Fermi energy levels shifted to zero). **d** Structural projections along the [100] direction for stoichiometric $CeO_2$ and non-stoichiometric $CeO_{1.5}$; surfaces of identical electronic charge are highlighted with blue colour; the electronic density increases around the Ce ions, which are located near the oxygen vacancies, compared to the stoichiometric system. Large yellow and small red spheres represent Ce and O ions, respectively

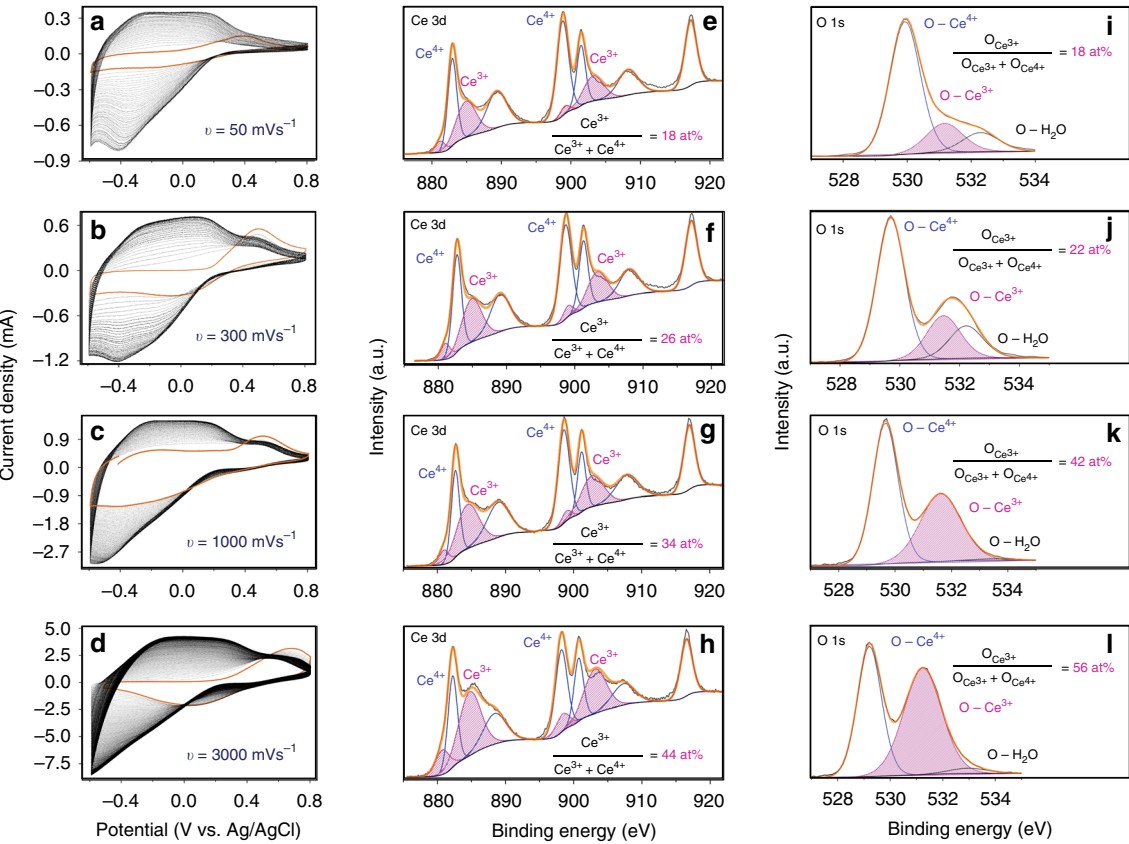

**Fig. 5** Proton-assisted control of oxygen vacancy formation during cyclic voltammetry. Cyclic voltammograms at scan rates: **a** 50 mV s$^{-1}$. **b** 300 mV s$^{-1}$. **c** 1000 mV s$^{-1}$. **d** 3000 mV s$^{-1}$. **e**–**h** Corresponding XPS Ce 3$d$ spectra. **i**–**l** Corresponding XPS O 1$s$ spectra

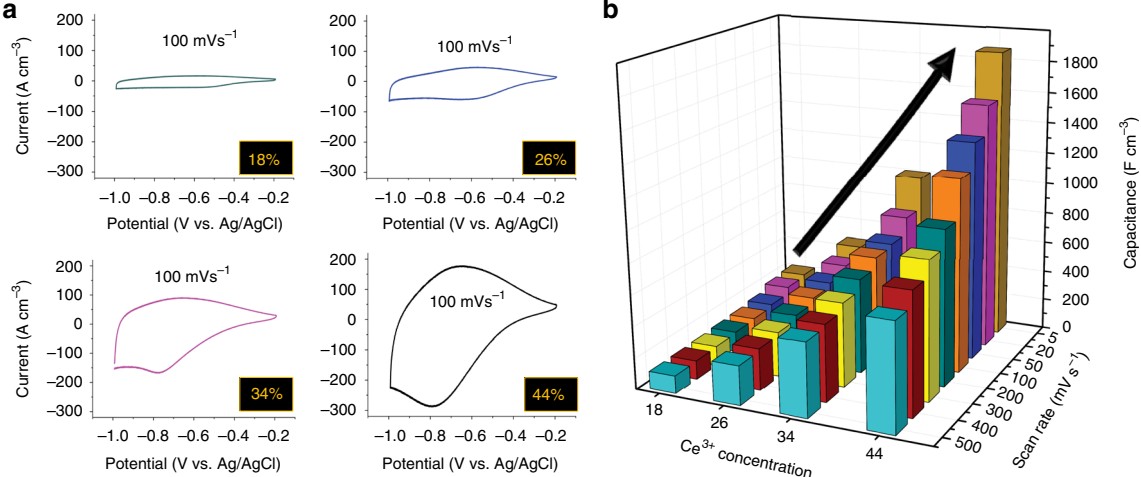

**Fig. 6** Effect of scan rate on $Ce^{3+}$ concentration and capacitance of thin films. **a** Pseudocapacitance-related cyclic voltammograms of $CeO_{2-x}$ deposited at scan rates of 50 mV s$^{-1}$ (green), 300 mV s$^{-1}$ (blue), 1000 mV s$^{-1}$ (purple) and 3000 mV s$^{-1}$ (black) (percentage in box is [$Ce^{3+}$]). **b** Three-dimensional plot showing volumetric capacitances at scan rates of 5–500 mV s$^{-1}$ for thin films with [$Ce^{3+}$] of 18, 26, 34 and 44 at%

creation of a thinner layer of $CeO_{2-x}$. The application of a large reduction current to the layer during each reverse scan leads to the creation of high [$V_O^{\bullet\bullet}$] effectively throughout the entire film. These conclusions are supported experimentally by the Ce 3$d$ XPS results in Fig. 5e–h, which show increasing [$Ce^{3+}$] from 18 at% to 44 at% at scan rates of 50 mV s$^{-1}$ to 3000 mV s$^{-1}$, respectively; the film thicknesses for 100 cycles were in the range ~25 nm to ~70 nm (i.e., ~1–3 atomic layers per cycle). Further, the XPS results for the O 1$s$ XPS spectra in Fig. 5i–l show that the fractional [$O$]$_{Ce^{3+}}$ increases from 18 at% to 56 at% at scan rates of 50 to 3000 mV s$^{-1}$, respectively, which also confirms that higher scan rates lead to the generation of higher [$V_O^{\bullet\bullet}$]. Additionally, the XPS results are confirmed by photoluminescence (PL) spectroscopy of the thin films, the results of which are given in Supplementary Fig. 18.

Figure 6a shows increasing areas of the pseudocapacitance-related cyclic voltammograms as a function of [$Ce^{3+}$] at the associated scan rates (electrochemical performances are given in Supplementary Fig. 17). The electrochemical performance data are summarised in Fig. 6b, which shows that increasing the [$Ce^{3+}$] (and corresponding [$V_O^{\bullet\bullet}$]) by ~2.5 times results in an eightfold increase in the volumetric capacitance.

The effects of variation of the cycle number at a constant scan rate on the change in film thickness also were considered (XPS, TEM, AFM and TOFSIMS results are in Supplementary Note 11). The data show that the film thickness is directly proportional to the cycle number. In contrast, increasing the cycle number decreased the [$V_O^{\bullet\bullet}$], as reflected by the [$Ce^{3+}$]. These converse data suggest the reason for both the maximum in the volumetric capacitance in Supplementary Fig. 23 as well as the dominant role of the [$V_O^{\bullet\bullet}$]. Finally, improvements in the volumetric capacitance can be achieved by increasing the [$V_O^{\bullet\bullet}$] to the theoretically maximal 25% by increasing the scan rate from that used in our work by up to a possible two orders of magnitude.

## Discussion

In summary, the present work reports a straightforward approach to fabricate ultrathin films with high [$V_O^{\bullet\bullet}$] and precisely controlled thickness using near-room-temperature cyclic voltammetry at large cycle numbers and high scan rates. Critically, by leveraging the slow kinetics of the cerium redox reactions and the PAOVC mechanism, layer-by-layer deposition of highly reduced $CeO_{2-x}$ with homogeneous volumetric distributions of $V_O^{\bullet\bullet}$ was

achieved. Further, rapid cyclic deposition was shown to create volumetric $V_O^{\bullet\bullet}$ with long-term stability, which has led to the observation of pseudocapacitive behaviour in $CeO_{2-x}$. This behaviour is attributed to simultaneous intracrystallite electron conduction owing to high volumetric [$V_O^{\bullet\bullet}$] and rapid inter-crystallite proton transfer activated by adsorbed water molecules. These nanostructures, which can be engineered for superior performance using scan rates up to two orders of magnitude greater than those used in the present work, exhibit the highest volumetric capacitance of any ultrathin film to date (Supplementary Table 12). The present work shows that the PAOVC method has the potential to be applied to other MOs for a range of applications in energy and the environment, e.g., gas sensing, oxygen storage and energy storage.

## Methods

**Electrochemical cell preparation.** The electrodeposition was performed using a classical three-electrode configuration system[48,49]. Fluorine-doped tin oxide on glass (FTO; Wuhan Geao Scientific Education Instrument, China; 2.0 cm × 1.5 cm) with a film resistivity of ~18 Ω, platinum (Basi Inc., Indiana, USA, coil $L$ = 23 cm, wire $D$ = 0.5 mm), and Ag/AgCl (Basi Inc., West Lafayette, IN, USA) were used as the working, counter and reference electrodes, respectively. All potentials were based on the Ag/AgCl reference electrode unless otherwise stated. Prior to electrodeposition, the FTO substrate was cleaned by ultrasonication in acetone and ethanol for 5 min each, followed by activation by immersion (1 cm) in 45% nitric acid for 2 min and drying with compressed nitrogen. The active surface area of the FTO hence was 1.5 cm$^2$. The potentiostat used was from EZstat Nuvant Systems, Inc., with a resolution of 300 µV and 3 nA at the ±100 µA range. During deposition, cyclic voltammetry was applied at varying scan rates, total times, potentials and currents at different pH values and temperatures. The resultant films were rinsed with deionised (DI) water and dried at room temperature prior to further analysis.

**Synthesis of ultrathin films.** Ultrathin films of $CeO_{2-x}$ were deposited electrochemically on FTO substrates using cyclic voltammetry at scan rates in the range 50–3000 mV s$^{-1}$ and cycle numbers in the range 50–2000 using an aqueous electrolyte. The electrolyte was synthesised by mixing 0.05 M Ce(NO$_3$)$_3$·6H$_2$O and 0.05 M Ca(C$_2$H$_3$O$_2$)$_2$ in DI water (resistance 18.2 MΩ cm); 1 M NaOH was used to adjust the pH to value of 6.

**Characterisation.** Mineralogical data for the films was obtained using a Philips X'Pert Multipurpose X-Ray Diffractometer (MPD; Almelo, Netherlands, CuKα radiation [0.15405 nm], 20°–80° 2$\theta$, step size 0.02° 2$\theta$, scanning speed 5.5° 2$\theta$ min$^{-1}$). The peaks were analysed using X'Pert High Score Plus software. These data were supplemented by laser Raman microspectra (Raman; Renishaw in Via Raman microscope, Gloucestershire, UK; beam diameter 1.5 µm), which was equipped with a 35 mW helium-neon green laser (514 nm), in the range 200–800 cm$^{-1}$. The spectra were fitted and calibrated using Renishaw WiRE 4.3 software. Electron micrographs of the thin-film nanostructures were obtained by high-resolution transmission electron microscopy (HRTEM; Philips CM 200, Eindhoven, the

Netherlands) and field emission gun scanning/transmission electron microscopy (FEG-STEM; JEOL JEM-F200 MultiPurpose FEG-STEM, Tokyo, Japan). The data were analysed using Thermo Scientific Avantage software. Dual-beam focused ion milling (FIB; FEI xT Nova NanoLab 200, Hillsboro, OR, USA) was used to prepare specimens for TEM imaging analyses. Mechanical polishing in combination with a precision ion polishing system (PIPS; Gatan PIPS II Model 695, Pleasanton, CA, USA) were used for sample preparation for TEM, EELS and HAADF analyses. Elemental mapping was done by TEM (Philips CM200, Eindhoven, Netherlands) equipped with EDS. Scanning electron microscopy images were obtained by SEM (FEI Nova NanoSEM; secondary electron emission; accelerating voltage 5 kV, Hillsboro, OR, USA). Surface chemical analyses were done by X-ray photoelectron spectroscopy (XPS; Thermo Fisher Scientific ESCALAB 250Xi spectrometer, 13 kV, 12 mA, spot size 500 μm, Loughborough, Leicestershire, UK). These data were supplemented by electron energy loss spectroscopy (EELS) spectra and associated high-angle annular dark-field (HAADF) images obtained by probe-corrected scanning transmission electron microscopy (STEM; JEOL JEM-ARM200F, Tokyo, Japan). To minimise reduction by the electron beam, the specimen was cooled in situ to liquid nitrogen temperature. The beam flux was reduced to the very low value of 15 pA to minimise the beam damage effects. Photoluminescence (PL) spectroscopy was done using a spectrofluorophotometer (RF-5301PC, Shimadzu, Kyoto, Japan). The thickness measurements were assessed by atomic force microscopy (AFM; Bruker Dimension Icon SPM, PeakForce Tapping mode, Billerica, MA, USA). A ScanAsyst-Air probe (Bruker AFM probes, Billerica, MA, USA) was installed in the AFM holder and used for all measurements. The samples were restrained on the stage using a slight vacuum. The scan area was set at 30 μm × 7.5 μm with an aspect ratio of 4; the pixel resolution was 512 samples/line (slow scan axis) and 128 line (fast scan axis), respectively, for the two dimensions. A slow scan rate of 0.195 Hz was used to ensure accuracy. The peak force was minimised to avoid sample deformation and the feedback gain settings were optimised accordingly. The thicknesses of the thin films were determined using the box option consisting of 128 lines, instead of using a single line.

Additionally, time-of-flight secondary ion mass spectrometry (TOFSIMS) was carried out using a TOF.SIMS 5 (ION-TOF GmbH, Munster, Germany) to confirm the thickness measurements. This spectroscopy was done using the spectrum imaging mode with sub-pixel scanning in operation. This ensured that, at all times during the acquisition, the beam was moving and the local flounce was minimised.

**First-principles calculations**. The PBEsol functional[50], as implemented in the VASP software package[51], was used. A "Hubbard-U" scheme[52], with $U = 3$ eV, was employed for superior treatment of the localised Ce 4f electronic orbitals (the adopted PBEsol + U set-up was confirmed to reproduce the experimental lattice parameter and band-gap of bulk $CeO_2$ as closely as possible). The "projector augmented wave" method was used to represent the ionic cores[53] by considering the following electrons to be valence: Ce: 5s, 5p, 6s, 5d and 4f; O: 2s, 2p. The wave functions were represented on a plane-wave basis truncated at 650 eV. For integrations within the Brillouin zone, Monkhorst-Pack k-point grids[54], with density equivalent to that of $16 \times 16 \times 16$ over the fluorite unit cell, were used. Geometry relaxations were done using a conjugate gradient algorithm that accommodated cell volume and cell shape variations; the geometry relaxations were halted once the forces on the atoms fell below 0.01 eV Å$^{-1}$. These technical parameters yielded zero-temperature energies that converged to within 0.5 meV per formula unit. In order to estimate accurate energy band gaps, the hybrid HSE06 exchange-correlation functional was used. Our spin-polarised density functional theory calculations were performed on a supercell containing 24 atoms for the stoichiometric system. The non-stoichiometric geometries were generated by successively removing oxygen atoms from the simulation box.

**Electrochemical measurements**. The electrochemical performance of the ultra-thin films was studied using a three-electrode system by cyclic voltammetry as a component of an electrochemical station (Ezstat Pro, Crown Point, IN, USA). The system included an Ag/AgCl electrode as reference electrode (Basi Inc., West Lafayette, IN, USA), Pt wire as described previously as counter electrode (Basi Inc., West Lafayette, IN, USA) and $CeO_{2-x}$ film on FTO substrate as working electrode in 1 M NaCl aqueous electrolyte; the pH was adjusted to 7 using a 1 M NaOH solution. Cyclic voltammetry was measured using a negative potential window in the range −1.0–0.2 V vs. Ag/AgCl electrode at room temperature for scan rates in the range 5–500 mV s$^{-1}$. In order to confirm that the total capacitances obtained could be attributed to the ultrathin $CeO_{2-x}$ films, the volumetric capacitance of bare FTO was determined to be insignificant (≤5 × 10$^{-8}$% of that of the $CeO_{2-x}$ capacitance).

## Data availability

All data are available within the manuscript and supplementary information. Further information can be acquired from the corresponding authors upon reasonable request.

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

## Acknowledgements

This work has been supported by the Australian Research Council (DP170104130). S.S. M. and E.A. acknowledge UPA and RTP, respectively, scholarship support. We are grateful for access to the characterisation facilities provided by the Mark Wainwright Analytical Centre, UNSW Sydney. We also are grateful to the National Computational Infrastructure (NCI), which is supported by the Australian Government, for permission to use Gaussian03.

## Author contributions

S.S.M. designed the project; undertook all of the syntheses and the majority of the characterisation, thermodynamic calculations and data analyses; he wrote the first draft of the manuscript and worked on all subsequent drafts. E.A. did the PIPS sample pre-paration, assisted in TEM and EELS imaging, and contributed to the data analysis; he also worked on all of the drafts of the manuscript. Y.Y. conducted AFM measurements and prepared TEM samples by FIB. P.K. contributed to the data analysis and revised all drafts of the manuscript. R.W. obtained the HAADF and line scanning results. X.L. undertook the PL spectroscopy measurements. R.K.N. contributed to the thermodynamic calculations. S.L. contributed to the TEM imaging for the thickness measurements. C.C. and Z.L. undertook the DFT calculations and wrote the relevant text. Y.W. assisted with the thickness measurements of the ultrathin films. N.L. contributed to obtaining ther-modynamic data by DFT calculation. C.C.S. worked on all subsequent drafts of the manuscript and supervised the overall project. All authors have provided their comments on the final version of the manuscript.

## Additional information

**Competing interests:** The authors declare no competing interests.

