## [Peer Review File · Nature Communications]

Reviewers' comments:

Reviewer #1 (Remarks to the Author):

I read through the whole manuscript, I have several concerns:

- 1 the authors should focus on the size effect of ceria, particularly in this topic. Some findings like "Size-dependent oxygen storage ability of nano-sized ceria" (Phys. Chem. Chem. Phys., 2013,15, 14414-14419) show that the size effect should be emphasized herein.
- 2 The calculations of Ce concentration were correlated with XPS data, I am not sure about its accuracy, if more direct evidences for this will be great.
- 3 As to the measured specific capacitance value of Ce-compounds, the authors may compare their data with available large values like Journal of Colloid and Interface Science, Volume 446, 15 May 2015, Pages 77-83; Journal of Colloid and Interface Science, Volume 416, 15 February 2014, Pages 172-176.
- 4 As to Fig.4c, I think it is hard to believe their data. I suggest the authors to recalculate their results with different sizes.

Reviewer #2 (Remarks to the Author):

This manuscript reports a kind of ultrathin CeO₂ material with oxygen vacancies. This material was prepared by using an electrodeposition technique and exhibited very high capacitance. The topic of the manuscript is helpful for the energy storages in the future and the electrochemical performance of the material is very good. However, there are some problems in the current manuscript and it must be revised before being accepted. The detailed problems are:

1. There have been few publications about the CeO₂ redox pseudocapacitive behavior, this manuscript is not the "first observation". For example:
APPLIED SURFACE SCIENCE, 2017, 449: 454
DALTON TRANSACTIONS, 2016, 45(36): 14352
So, the authors should revise the statements in the manuscript.
2. What does the prepared ultrathin CeO_{2-x} material look like? Like membranes which cover the FTO substrate layer by layer? Please provide some SEM images of the morphologies of the material.
3. What is the loading of the material? If the loading is very low, the electrochemical performance would be very high while the material is not promising actually.
4. In Fig2, the authors identified the peaks (Ox1, Ox2, Ox3, Re1, Re2 and Re3) in the CV curves, but the evidence is not enough. Please write the chemical reaction equations for each peak and confirm them. If the authors insist on the explanations of the peaks, please calculate the areas of every peak and discuss the variation trend of the areas.
5. The thicknesses of the films are obtained by TEM technique (in Supplementary Fig.12 and Fig. 15). However, the field of view of TEM is a very narrow. Moreover, the CeO_{2-x} films are not uniform (for example, Supplementary Fig.12 b is a typical image). The thicknesses of films should be measured by using other suitable techniques.
6. In Table 5 of Supplementary Section 5, the 2nd and 3rd reactions are unreasonable. Why is OH- the final product in an acidic surrounding? If the reaction equations are not true, the calculated results should be changed.

Reviewer 1

General Comment: I read through the whole manuscript, I have several concerns.

Response: The authors thank the reviewer for his/her constructive comments. The response to each of the comments is given below.

Comment 1: The authors should focus on the size effect of ceria, particularly in this topic. Some findings like “Size-dependent oxygen storage ability of nano-sized ceria” (Phys. Chem. Chem. Phys., 2013,15, 14414-14419) show that the size effect should be emphasized herein.

Response:

While we agree with reviewer’s point, the present work reports data for a narrow particle size range of 3-8 nm (Figures 1d, S5, S7 and S11), so investigation of a particle size effect is not possible. More broadly, the effect of this particular size range on the oxygen storage capacity of CeO₂ nanoparticles (*viz.*, nanocrystallites) merits comment. According to the experimental work reported by Hailstone RK *et al.*¹ in which the influence of particle size on the storage capacitance of CeO₂ nanoparticles is discussed (as is the case for the reference suggested by the reviewer²), the size range obtained in the present work is the optimal to achieve the highest oxygen storage capacity as it falls within a range that identified by these authors. The consistency of the crystallite size range obtained in the present work is due to the use of the low experimental temperature of 60°C during the cyclic voltammetry deposition. Nonetheless, variation in other experimental parameters allows the achievement of variable, controlled, and high densities of oxygen vacancies.

Further, the effect of particle size on the oxygen storage capacity of CeO₂ also depends on the exposed surface facets of single crystals (*e.g.*, octahedron consisting of (111) planes or cube consisting of (100) planes). In the present work, the polycrystalline nature of the thin films (Figures 1d, S5, S7 and S11) prevents such an analysis owing to the randomly exposed facets of the crystallites in the thin films, so measurement of Ce/O ratios for investigation on storage properties of the thin films cannot be done.

Accordingly, the following text and figure have been added to the main paper (Page 6) and supplementary materials (Page 17), respectively, in order to clarify the effect of crystallite size on the oxygen storage capacity of CeO₂.

Textual Modification:

Main paper (Page 6):

The high capacitances of the thin films can be attributed to the optimal crystallite sizes (3-8 nm; Supplementary Figure 11), which are nearly identical to the experimental particle size range reported³⁶ to exhibit the greatest surface Ce/O ratio and consequent high oxygen storage capacity (3-8 nm). In contrast, a sub-nanometre particle size has been projected by calculation to exhibit an optimal oxygen storage capacity³⁷.

Supplementary Material (Page 17):

Supplementary Figure 11. Thin film deposited at different scan rates 50-3000 mVs⁻¹. (a,b) HAADF images of ultrathin films deposited at 3000 mVs⁻¹ and 1000 mVs⁻¹, respectively. (c,d) HRTEM images, and (e,f) SAED patterns of ultrathin film deposited at 300 mVs⁻¹ and 50 mVs⁻¹, respectively.

Comment 2: The calculations of Ce concentration were correlated with XPS data, I am not sure about its accuracy, if more direct evidences for this will be great.

Response:

The authors would like to thank the reviewer for raising this important matter. The general aim of measuring Ce content (Ce^{4+} or Ce^{3+}) is for either quantitative or qualitative analysis of oxygen vacancies, as is the case for the present study. This work reports relevant quantitative data from XPS and supporting qualitative data from electron energy loss spectroscopy (EELS), line scanning of HAADF images, and laser Raman microspectroscopy. XPS analysis is the most-commonly used quantitative technique for measuring Ce and consequent oxygen vacancy concentration^{3, 4, 5}. Nonetheless, in line with the reviewer's suggestion, photoluminescence (PL) spectroscopy has been used to obtain comparative data indicating Ce^{3+} content change in the ultrathin films. This approach is used frequently for qualitative analysis of defect densities in CeO_2 materials.^{6, 7} The emission intensity in the PL spectra derives from charge-carrier (electron/hole) recombination, thereby suggesting the reduction in concentrations of mobile defects from the different mechanisms (*e.g.*, $\text{Ce}^{4+} \rightarrow \text{Ce}^{3+}$ redox, O vacancy formation, and exciton generation); this reduction results from recombination, which is revealed by PL peak intensity increase. The obtained PL data are described and incorporated in Supplementary Section 10 (Page 21), also shown below:

Textual Modification:**Supplementary Material (Page 21):**

Photoluminescence spectroscopy (PL), which is used commonly to evaluate relative charge carrier concentrations ($\text{Ce}^{4+} \rightarrow \text{Ce}^{3+}$ redox, O vacancy formation, and electron/hole pair generation)^{21,22}, also was conducted in the present work in order to confirm that cyclic voltammetry deposition can be used to control defect concentrations in CeO_{2-x} . The emission intensity in the PL spectra derives from charge-carrier (electron/hole) recombination, thereby suggesting reduction in the concentrations of mobile defects^{21,22} as the peak intensities increase.

Figure 18. PL spectra of ultrathin films prepared at different scan rates (100 cycles at 50-3000 mVs^{-1}) and cycle numbers (50-500 at 3000 mVs^{-1}).

Comment 3: As to the measured specific capacitance value of Ce-compounds, the authors may compare their data with available large values like Journal of Colloid and Interface Science, Volume 446, 15 May 2015, Pages 77-83; Journal of Colloid and Interface Science, Volume 416, 15 February 2014, Pages 172-176.

Response:

The authors thank the reviewer for providing the references, which show high capacity of cerium oxide materials as supercapacitors. Accordingly, the authors have consulted the references cited by the reviewer^{8, 9}. Owing to the ultrathin nature of the films in the present work, all of the capacitances reported are on the basis of volume and area of each electrode (*i.e.*, volumetric and areal capacitances). These data by nature are not comparable to those reported by others for the gravimetric capacitances of bulk samples¹⁰, as reported in the two suggested papers. By convention, the capacitances of ultrathin films of very low masses (ng- μg scale mass) are reported on the basis of volume or area^{11, 12, 13, 14}. This is due to the fact that the low masses of the ultrathin films greatly inflate the specific capacitance¹⁰, which does not reflect the actual capacitance. In contrast, the capacitances of nanoparticles (mg scale mass) are reported on the basis of mass (*i.e.*, gravimetric capacitance). Nonetheless, the authors believe that the very high capacitance reported in the suggested references is an excellent example of why CeO_2 materials should be explored further for energy storage materials. The high values reported for gravimetric capacitance draw attention to

the excellent performance of CeO₂, regardless of the method of measurement, and accordingly the following text has been added to the Introduction of the main paper (Page 1):

Textual Modification:

Main paper (Page 1):

Although intercalation pseudocapacitance has not been observed to date in CeO₂, very high specific capacitances of this material have been reported^{17, 18}, indicating its potential for energy storage applications.

Comment 4: As to Fig.4c, I think it is hard to believe their data. I suggest the authors to recalculate their results with different sizes.

Response:

In regard to the reviewer's drawing attention to the importance of particle size reported in suggested reference², which is cited in the revised version of the manuscript, some points of clarification concerning the computational methodology are merited. Also, the reviewer's comment has been addressed by modifying the manuscript to include textual changes, clarifications, and new numerical data have been generated.

In the present work, the main aim of the density functional theory (DFT) calculations is to provide physical insights into how the structural and electronic properties of bulk ceria change in the presence of oxygen vacancies. The DFT energy band gaps (E_g) reported in Figure 4c are for bulk stoichiometric and non-stoichiometric systems, so both systems must be periodic along the three Cartesian directions. Since the system boundaries are placed at infinity, by definition, nanosize effects cannot be considered in the simulations.

Further, electronic band structure calculations incorporating nano-size effects cannot be done using first-principles DFT because, for the electronic band structure properties of a system to be well defined, the system must be periodic along at least one direction. If this is not the case, then the Bloch theorem, on which the theory of electronic band structure of crystals is based, is not valid. Therefore, the electronic band structures of nanoparticles in principle are not well defined since they

are zero-dimensional and hence lack infinite periodicity. Further, in the present work, faceted ceria nanoparticles were not produced; instead ceria thin films (*i.e.*, periodic along two Cartesian directions) of ultrafine crystallites with randomly oriented faceting were synthesised.

However, in order to probe the role of nano-size effects on the DFT simulations, bearing in mind the preceding comments, new DFT simulations designed to replicate the features indicated in Figures 2 and 3 in the suggested reference². These data reveal that 2% expansion or contraction of the Ce-O bond lengths in stoichiometric and non-stoichiometric CeO_{2-x} (as done in the cited publication) results in alteration of the E_g by only ≤3%, confirming that there is little size effect on the electronic band structure. The following text and figure have been added to Supplementary Section 9 (Pages 17-18).

Textual Modification:

Supplementary Material (Pages 17-18):

Section 9. Effect of crystallite size on calculated band gap energy

It is possible that there may be nano-size effects on the estimation of the E_g in ceria, as suggested by atomic packing considerations of Ce-O bond lengths of nanoparticles¹⁹. However, such size effects cannot be simulated directly by DFT since the electronic band structure of a system requires periodicity in at least one dimension for the Bloch theorem to be valid and nanoparticles are zero-dimensional and hence lack periodicity. However, potential size effects on the E_g have been examined by the DFT calculations by considering 2% expansion or 2% contraction of the Ce-O bond lengths that correspond to analogous calculations based on the presence and absence of ligands in the growth environment¹⁹. The resultant DFT calculations for stoichiometric CeO₂ (Figure 12) reveal that 2% expansion or contraction in the Ce-O bond lengths leads to an almost negligible 2% reduction or 3% increase in the E_g, respectively. For non-stoichiometric CeO_{2-x}, 2% expansion or contraction results in only a 3% decrease or 2% increase in the E_g, respectively. In effect, if there is any size effect on the E_g corresponding to the preceding conditions¹⁹, it is not significant.

Supplementary Figure 12. Influence of potential nano-size effects on calculated energy band gap of stoichiometric and non-stoichiometric bulk ceria. Calculated spin-polarised total density of states showing spin-up (positive) and spin-down (negative) components; the Fermi energy level has been shifted to zero in all the cases.

Reviewer 2

General Comment: This manuscript reports a kind of ultrathin CeO₂ material with oxygen vacancies. This material was prepared by using an electrodeposition technique and exhibited very high capacitance. The topic of the manuscript is helpful for the energy storages in the future and the electrochemical performance of the material is very good. However, there are some problems in the current manuscript and it must be revised before being accepted.

Response: The authors thank the reviewer for his/her constructive comments. The response to each of the comments are given in below:

Comment 1: There have been few publications about the CeO₂ redox pseudocapacitive behaviour, this manuscript is not the “first observation”. For example:

APPLIED SURFACE SCIENCE, 2017, 449: 454

DALTON TRANSACTIONS, 2016, 45(36): 14352

So, the authors should revise the statements in the manuscript.

Response:

The authors appreciate this correction. All references to “first observation for redox pseudocapacitive behaviour” have been removed.

Comment 2: What does the prepared ultrathin CeO_{2-x} material look like? Like membranes which cover the FTO substrate layer by layer? Please provide some SEM images of the morphologies of the material.

Response:

Low- and high-resolution SEM images of the surfaces of coated and uncoated FTO substrates (scan rates 3000, 1000, 300, 50 mVs⁻¹) have been added to Supplementary Section 3 as Supplementary

Figure 4 (Page 9), as shown below. The very limited thicknesses of the ultrathin films allow the underlying FTO morphology to be observed; the acuity of the FTO grains is diminished with the thickest film (50 mVs^{-1}). Additionally, imaging of the ultrathin films is provided in the form of AFM profiles, which are given in the Response to Comment 5.

Textual Modification:

Supplementary Material (Page 9):

Supplementary Figure 4. SEM images of bare FTO substrate and CeO_{2-x} ultrathin films (100 cycles) deposited at scan rates of 3000, 1000, 300, and 50 mVs^{-1} (left-to-right sequence of figures is based on increasing thickness).

Comment 3: What is the loading of the material? If the loading is very low, the electrochemical performance would be very high while the material is not promising actually.

Response:

The loads of the ultrathin films were in the range $10\text{-}50 \mu\text{gcm}^{-2}$, which covers a wide range from low to high masses compared to a great number of ultrathin films reported previously, which were in the range $0.3\text{-}80 \mu\text{gcm}^{-2}$ ^{11, 12, 13, 14, 15, 16, 17}. In fact, the issue of how the capacitance for materials of different classifications (dimensionality and mass) should be evaluated and subsequently reported is an important controversial concept, which was the focus of a recent paper in Science¹⁰. The method of capacitance measurement is critical, especially when working with ultrathin films with very limited mass; this also is the case with hybrid materials containing graphene. As the reviewer noted,

for these materials, it is recommended not to report the capacitance based on the mass of the film but, instead, it should be reported based on the volume and/or area of the electrode materials. Therefore, owing to the potential to imply artificially high gravimetric capacitance, which is unrealistic relative to the real electrochemical performance of the films, all of the capacitance values reported in the present work are based on volume and surface area of the CeO_{2-x} electrode, in agreement with previous reports^{13,18, 19, 20}.

Comment 4: In Fig2, the authors identified the peaks (Ox1, Ox2, Ox3, Re1, Re2 and Re3) in the CV curves, but the evidence is not enough. Please write the chemical reaction equations for each peak and confirm them. If the authors insist on the explanations of the peaks, please calculate the areas of every peak and discuss the variation trend of the areas.

Response:

As the reviewer requested, all the peaks have been identified and the corresponding chemical reactions are given (Supplementary Section 6; Tables 5-7). Additionally, all of the peak areas, which correspond to the charge contribution to the reaction, have been deconvoluted using the Gaussian method and subsequently discussed (Supplementary Section 6; Table 8). These equations, area calculations, and discussion are also provided below. Owing to the challenging nature of thermodynamic calculations at the experimental temperature (60°C) and the high scan rates (as noted by Bockris²¹ and Matsuda²²) peaks were identified on the basis of analogous thermodynamic calculations for room temperature and slow scan rates. Also, the identification of the Ox₁ & Re₁ peaks and Ox₂ & Re₂ peaks is discussed comprehensively in Supplementary Sections 1 and 7.

Textual Modification:

Supplementary Material (Pages 12-14):

Section 6. Chemical identification and areal calculation of peaks observed in cyclic voltammetry deposition of CeO_{2-x} thin films

In this section, the chemical reactions associated with the redox peaks observed in cyclic voltammetry deposition of CeO_{2-x} thin films (Figure 2 in main text) are identified and discussed in

details. The oxidation of Ce^{3+} (Ox_1) occurs at $E = 0.47 \text{ V vs Ag/AgCl}$ followed by reduction (Re_1) at $E = -0.02 \text{ V vs Ag/AgCl}$. The accuracy of the identified peaks was confirmed over a wide range of pH values, the results of which are discussed comprehensively in Supplementary Section 1. Owing to the irreversibility of the redox reaction (Ox_1 and Re_1), as shown in Supplementary Section 1, some of the Ce(OH)_4 redissolves as soluble Ce^{3+} during reduction (Re_1), which is confirmed by the lower current density of Re_1 relative to that of Ox_1 ((anodic peak current)/(cathodic peak current) or $I_{pa}/I_{pc} < 1$). These reactions and corresponding calculations are given in Table 5.

Supplementary Table 5. Chemical reactions attributed to $\text{Ox}_1 - \text{Re}_1$ peaks and their corresponding energy calculation.

$\text{Ox}_1 - \text{Re}_1$ (Oxidation of Ce^{3+})		Calculated ΔG^0 (kJ/mol)	ΔG^0 (kJ/mol)
1	$\text{Ce}^{4+} + e \rightarrow \text{Ce}^{3+}$	-168.20	$\Delta G_{\text{Ce}^{4+}}^0 = -503.80$ $\Delta G_{\text{Ce}^{3+}}^0 = -672.00$
2	$4\text{H}_2\text{O} \rightarrow 4\text{H}^+ + 4\text{OH}^-$	+319.60	$\text{Log } K_w = 14.00$
3	$\text{Ce(OH)}_{4(\text{pt})} + 4\text{H}^+ \rightarrow \text{Ce}^{4+} + 4\text{H}_2\text{O}$	-0.02	$\Delta G_{\text{Ce(OH)}_{4(\text{pt})}}^0 = -1450.25$
Total	$\text{Ce(OH)}_{4(\text{pt})} + e \leftrightarrow \text{Ce}^{3+} + 4\text{OH}^-$	$E = -1.56 + 0.24 \text{ p(OH)} - 0.06 \log \frac{[\text{Ce}^{3+}]}{[\text{Ce(OH)}_{4(\text{pt})}]}$ $E (\text{pH} = 6) = +0.16 \text{ V vs Ag/AgCl}$ $E_{1/2} = +0.22 \text{ V vs Ag/AgCl}$	

The remaining Ce(OH)_4 deposited on the substrate transforms rapidly to CeO_2 (as indicated in the Pourbaix diagram in Figure 3(d)). As given in Table 6, the reduction of as-prepared CeO_2 occurs at $E = -0.13 \text{ V vs Ag/AgCl}$. This is followed by partial oxidation (annihilation of oxygen vacancies) by the following oxidation reaction (Ox_2) occurring at $E = +0.07 \text{ V vs Ag/AgCl}$. This is further confirmed in Supplementary Section 7.

Supplementary Table 6. Chemical reactions attributed to $\text{Ox}_2 - \text{Re}_2$ peaks and their corresponding energy calculation.

$\text{Ox}_2 - \text{Re}_2$ (Reduction of CeO_2)		ΔG^0 (kJ/mol)
1	$\text{CeO}_2 + 2xe^- \rightarrow \text{CeO}_{2-x} + x\text{O}^{2-}$	+165.00
2	$\text{O}^{2-} + \text{H}^+ \rightarrow \text{OH}^-$	-157.20
3	$\text{CeO}_2 + x\text{H}^+ + 2e^- \rightarrow \text{CeO}_{2-x} + x\text{OH}^-$	$E = +0.08 + 0.03 \text{ p(OH)} - 0.03 \text{ pH} - 0.03 \log \frac{[\text{CeO}_{2-x}]}{[\text{CeO}_2]}$ $E (\text{pH} = 6) = -0.06 \text{ V vs Ag/AgCl}$ $E_{1/2} = -0.03 \text{ V vs Ag/AgCl}$

Owing to the formation of oxygen vacancies within the bulk of the deposited CeO_{2-x} , the insertion/disinsertion of protons commences during the reduction (Re_3) and oxidation (Ox_3) reactions, as given in Table 7. Although these phenomena have not been observed previously for

CeO₂-based materials, they have been reported for a few metal oxides (e.g., WO₃^{11,12}, RuO₂^{13,14}, and MnO₂¹⁵).

Supplementary Table 7. Chemical reactions attributed to Ox₃ – Re₃ peaks and their corresponding energy calculation.

Ox₃ – Re₃ (H⁺ Intercalation/Disintercalation)	
Re₃	$Ce_{1-x}^{4+}Ce_x^{3+}O_{2-y} + \alpha e^- + \beta H^+ \rightarrow Ce_{1-(x+\alpha)}^{4+}Ce_{x+\alpha}^{3+}O_{2-(y+\delta)}OH_{(\beta+y)}$
Ox₃	$Ce_{1-(x+\alpha)}^{4+}Ce_{x+\alpha}^{3+}O_{2-(y+\delta)}OH_{(\beta+y)} \rightarrow Ce_{1-x}^{4+}Ce_x^{3+}O_{2-y} + \alpha e^- + \beta H^+$

The charge contributions of each pair of reactions were calculated using Gaussian fitting. As can be seen from the corresponding values in Table 8, the contribution percentage of Ox₁ – Re₁ decreases by increase the cycle number, while the actual values have insignificant decrease over cycling. Such phenomenon can be ascribed to continuous growth of the ultrathin film over cycling. Nonetheless, there are significant growth for both Ox₂ – Re₂ and Ox₃ – Re₃ by increase the cycle numbers. The growth in the peak area of Ox₂ – Re₂ can be ascribed to the total increase in oxygen vacancy concentration formed within bulk of the ultrathin film. Consistent with thickening of the CeO₂ film and increasing the amount of oxygen vacancies, the volume of hydrogen insertion/disinsertion reactions increase, resulting in having the highest areal contribution in the cyclic voltammogram.

Supplementary Table 8. Charge contribution and area calculation for identified peaks.

	Cycle 1	Cycle 25	Cycle 50
Redox reaction	Ox₁ – Re₁ & Re₂	Ox₁ – Re₁, Ox₂ – Re₂, Ox₃ – Re₃	Ox₁ – Re₁, Ox₂ – Re₂, Ox₃ – Re₃
Shared peak area for Ox ₁ /Re ₁ (%)	29.4 x 10 ⁻⁵ C (77%)	26 x 10 ⁻⁵ C (25%)	20.9 x 10 ⁻⁵ C (16%)
Shared peak area for Ox ₂ /Re ₂ (%)	8.6 x 10 ⁻⁵ C (23%) (only Re ₂)	38 x 10 ⁻⁵ C (37%)	44.9 x 10 ⁻⁵ C (36%)
Shared peak area for Ox ₃ /Re ₃ (%)	0 x 10 ⁻⁵ C (0%)	37 x 10 ⁻⁵ C (38%)	55.8 x 10 ⁻⁵ C (48%)
Total peak area	38 x 10 ⁻⁵ C	101 x 10 ⁻⁵ C	122 x 10 ⁻⁵ C

Comment 5: The thicknesses of the films are obtained by TEM technique (in Supplementary Fig.12 and Fig. 15). However, the field of view of TEM is a very narrow. Moreover, the CeO_{2-x} films are not uniform (for example, Supplementary Fig.12 b is a typical image). The thicknesses of films should be measured by using other suitable techniques.

Response:

Following the reviewer's request, the thicknesses of the films deposited were characterised by two additional methods:

- 1) AFM analyses were used to measure the thicknesses of the ultrathin films by scanning the cross-sectional area ($21 \mu\text{m}^2$) between film and substrate.
- 2) Time-of-flight secondary ion mass spectrometry (TOFSIMS) was used to confirm the thicknesses (scanning surface area $900 \mu\text{m}^2$). In order to highlight these data, 3-dimensional imaging was done. This allowed the areas of the CeO_{2-x} ultrathin films to be differentiated completely from the FTO substrates by elemental mapping.

The AFM and TOFSIMS data are given in Supplementary Section 10 (Supplementary Figure 16) and 11 (Supplementary Figure 21), as shown below. These data confirm the previously reported thicknesses obtained by TEM images.

Textual Modification:

Supplementary Material (Page 20):

Supplementary Figure 16. Thicknesses of CeO_{2-x} ultrathin films synthesised at constant cycle number 100 and different scan rates: (a) 50, (d) 300 (g) 1000, (j) 3000 mVs^{-1} : (a, d, g, j) AFM images at cross sectional regions between ultrathin film and FTO substrate; (b,e,h,k) height profile plots of corresponding AFM images; (c,f,i,l) 2D and 3D SIMS elemental images indicating thicknesses of CeO_{2-x} ultrathin films.

Supplementary Material (Page 23):

Supplementary Figure 21. Thicknesses of CeO_{2-x} ultrathin films synthesised at constant scan rate 3000 mVs^{-1} and different cycle numbers: (a) 50, (d) 100 (g) 200, (j) 500: (a, d, g, j) AFM images at cross sectional regions between ultrathin film and FTO substrate; (b,e,h,k) height profile plots of corresponding AFM images; (c,f,i,l) 2D and 3D SIMS elemental images indicating thicknesses of CeO_{2-x} ultrathin films.

Comment 6: In Table 5 of Supplementary Section 5, the 2nd and 3rd reactions are unreasonable. Why is OH^- the final product in an acidic surrounding? If the reaction equations are not true, the calculated results should be changed.

Response:

The authors thank the reviewer for highlighting the necessity of clarifying the apparent contradiction whereby OH^- is a product under acidic conditions. In order to confirm and clarify the effect of pH, differentiating calculations and experiments were done. This additional work is discussed in Supplementary Section 7, as given below:

Textual Modification:

Supplementary Material (Pages 15-16):

Since the formation of OH⁻ in an acidic environment might seem counter-intuitive, we have conducted further thermodynamic calculation. As shown in Table 10, oxygen vacancy formation in a relatively strong acidic environment (*e.g.*, pH ≤ 5.5) rich in hydrogen ions leads to the local formation of water¹⁷. In such a condition, the thermodynamic calculations (Table 10) show that the required Gibbs free energies (and corresponding E_{1/2}) for oxygen vacancy formation shift to values less than -1.26 eV vs. Ag/AgCl for pH ≤ 5.5. Such a high energy requirement for an aqueous system falls within the hydrogen-evolution region, as confirmed by the corresponding Pourbaix diagram (2H⁺ + 2e → H₂)^{1,3}. This is consistent with the cyclic voltammetry results (Supplementary Figure 8c), where CeO₂ film deposition occurred at pH = 5.5 but there was no cyclic voltammetry peak from oxygen vacancy formation at values of approximately -0.02 eV vs. Ag/AgCl. Cycling at higher acidic pH values of 4.5 and 5.0 also showed the absence of an oxygen vacancy formation peak.

Supplementary Table 10. Thermodynamic calculation of oxygen vacancy formation reaction at different pH (5.0-5.8).

No.	Reaction	ΔG ⁰ (eV)	
1	CeO ₂ + 2xe ⁻ → CeO _{2-x} + xO ²⁻	+1.71 (~1.20 to ~2.30)	
2	xO ²⁻ + 2xH ⁺ → xH ₂ O	-2.45	
3	CeO ₂ + 2xH ⁺ + 2xe ⁻ → CeO _{2-x} + xH ₂ O	$E = -0.74 - 0.06 \text{ pH} - 0.06 \log \frac{[\text{CeO}_{2-x}]}{[\text{CeO}_2]}$	
Gibbs Free Energy of V ₀ ^{**} Formation through H ₂ O Formation		pH = 4.5	ΔG _{V₀} = -1.20 eV
		pH = 5	ΔG _{V₀} = -1.23 eV
		pH = 5.5	ΔG _{V₀} = -1.26 eV

In contrast, as shown in Table 11, oxygen vacancy formation in a weakly acidic environment (*e.g.*, pH ≥ 6) less rich in hydrogen ions leads to the formation of OH⁻ ions, thus making the local environment essentially basic. In the presence of CeO_{2-x}, the amount of H⁺ in basic solutions is much lower than acidic solutions, so the extent of reaction between as-produced O²⁻ with two H⁺, leading to H₂O formation, is reduced significantly. Therefore, the formation of OH⁻ is favoured over that of H₂O¹⁷, as confirmed by the calculations given in Table 11. This is consistent with the cyclic voltammetry results shown in Figure 8 (Supplementary) and Figure 2 (main text), where CeO₂ film deposition occurred at pH = 6.0 and was followed by the oxygen vacancy formation peak at E_{1/2} = ~-0.03 eV vs. Ag/AgCl. Cycling at less acidic (*i.e.*, more basic) pH values of 6.5 and 7.0 also showed the presence of an oxygen vacancy formation peak. In effect, a minimal pH of 6.0 is required in order to form an oxygen vacancy at accessible energies of approximately -0.1 eV vs. Ag/AgCl, depending on pH of the environment.

Supplementary Table 11. Thermodynamic calculation of oxygen vacancy formation reaction at different pH (6.0-7.0).

No.	Reaction	ΔG^0 (eV)
4	$CeO_2 + 2xe^- \rightarrow CeO_{2-x} + xO^{2-}$	+1.71 (~1.20 to ~2.30)
5	$xO^{2-} + xH^+ \rightarrow xOH^-$	-1.63
6	$CeO_2 + xH^+ + 2e^- \rightarrow CeO_{2-x} + xOH^-$	$E = +0.08 + 0.03 p(OH) - 0.03 pH - 0.03 \log \frac{[CeO_{2-x}]}{[CeO_2]}$
Gibbs Free Energy of $V_O^{\bullet\bullet}$ Formation through OH^- Formation		pH = 6.0
		$\Delta G_{V_o} = -0.06$ eV
		pH = 6.5
	$\Delta G_{V_o} = -0.09$ eV	
	pH = 7.0	
	$\Delta G_{V_o} = -0.12$ eV	

References

1. Hailstone, R.K. *et al.* A Study of Lattice Expansion in CeO₂ Nanoparticles by transmission electron microscopy. *J. Phys. Chem. C*. **113**, 15155-15159 (2009).
2. Sun, C. & Xue, D. Size-dependent oxygen storage ability of nano-sized ceria. *Phys. Chem. Chem. Phys.* **15**, 14414-14419 (2013).
3. Barth, C. *et al.* A perfectly stoichiometric and flat CeO₂(111) surface on a bulk-like ceria film. *Sci. Rep.* **6**, doi:ARTN 2116510.1038/srep21165 (2016).
4. Naganuma, T. Traversa, E. Stability of the Ce³⁺ valence state in cerium oxide nanoparticle layers. *Nanoscale*. **4**, 4950-4953 (2012).
5. Stetsovych, V. *et al.* Epitaxial cubic Ce₂O₃ films via Ce–CeO₂ interfacial reaction. *J. Phys. Chem. Lett.* **4**, 866-871 (2013).
6. Channei, D. *et al.* Photocatalytic degradation of methyl orange by CeO₂ and Fe–doped CeO₂ films under visible light irradiation. *Sci. Rep.* **4**, 5757, doi:10.1038/srep05757 (2014).
7. Tamizhdurai, P. *et al.* Environmentally friendly synthesis of CeO₂ nanoparticles for the catalytic oxidation of benzyl alcohol to benzaldehyde and selective detection of nitrite. *Sci. Rep.* **7**, 46372, doi:10.1038/srep46372 (2017).
8. Chen, K. & Xue, D. Water-soluble inorganic salt with ultrahigh specific capacitance: Ce(NO₃)₃ can be designed as excellent pseudocapacitor electrode. *J. Colloid. Interface. Sci.* **416**, 172-176 (2014).
9. Chen, K. & Xue, D. In-situ electrochemical route to aerogel electrode materials of graphene and hexagonal CeO₂. *J. Colloid. Interface. Sci.* **446**, 77-83 (2015).
10. Gogotsi, Y. & Simon, P. True performance metrics in electrochemical energy storage. *Science*. **334**, 917 (2011).
11. Xu, P. *et al.* Laminated ultrathin chemical vapor deposition graphene films based stretchable and transparent high-rate supercapacitor. *ACS Nano*. **8**, 9437-9445 (2014).
12. Wu, Z.-S. *et al.* Bottom-up fabrication of sulfur-doped graphene films derived from sulfur-annulated nanographene for ultrahigh volumetric capacitance micro-supercapacitors. *Journal of the American Chemical Society*. **139**, 4506-4512 (2017).
13. Wu, Z.-S. *et al.* Stacked-layer heterostructure films of 2D thiophene nanosheets and graphene for high-rate all-solid-state pseudocapacitors with enhanced volumetric capacitance. *Adv. Mater.* **29**, 1602960 (2017).
14. Zhu, Q. Xie, C. Li, H. Yang, C. Zeng, D. A novel planar integration of all-solid-state capacitor and photodetector by an ultra-thin transparent sulfated TiO₂ film. *Nano Energy*. **9**, 252-263 (2014).
15. Ghosh, A. *et al.* High pseudocapacitance from ultrathin V₂O₅ films electrodeposited on self-standing carbon-nanofiber paper. *Adv. Funct. Mater.* **21**, 2541-2547 (2011).
16. Borysiewicz, M.A. Wzorek, M. Myśliwiec, M. Kaczmarek, J. Ekielski, M. MnO₂ ultrathin films deposited by means of magnetron sputtering: relationships between process conditions, structural properties and performance in transparent supercapacitors. *Superlattices and Microstructures*. **100**, 1213-1220 (2016).
17. Hou, Y. Chen, L. Liu, P. Kang, J. Fujita, T. Chen, M. Nanoporous metal based flexible asymmetric pseudocapacitors. *J. Mater. Chem. A*. **2**, 10910-10916 (2014).
18. Zhou, J. *et al.* Ultrahigh volumetric capacitance and cyclic stability of fluorine and nitrogen co-doped carbon microspheres. *Nat. Commun.* **6**, 8503 (2015).
19. Wu, C. Z. *et al.* Two-dimensional vanadyl phosphate ultrathin nanosheets for high energy density and flexible pseudocapacitors. *Nat. Commun.* **4**, doi:ARTN 243110.1038/ncomms3431 (2013).
20. Feng, J. *et al.* Metallic Few-Layered VS₂ Ultrathin nanosheets: high two-dimensional conductivity for in-plane supercapacitors. *J. Am. Chem. Soc.* **133**, 17832-17838, doi:10.1021/ja207176c (2011).
21. Bockris, J. O. M. Reddy, A. K. N. & Gamboa-Aldeco, M. E. *Modern Electrochemistry 2A: Fundamentals of Electrode Processes*. (Springer US, 2007).

22. Matsuda H, Ayabe Y. Zur Theorie der Randles-Sevčičsches Kathodenstrahl-Polarographie. Zeitschrift für Elektrochemie, Berichte der Bunsengesellschaft für physikalische Chemie. **59**: 494-503, doi:10.1002/bbpc.19550590605 (1955).

Reviewers' comments:

Reviewer #1 (Remarks to the Author):

I read through all files presented by this work, much better than before. While I still have one more concern about their "First-Principles Calculations", something fundamental may be clarified if these authors can make this try, for example, the electrons as valence for Ce should be 4f, 5d, 6s, 6p. Even the bonding conditions for rare earth like Ce are complicated, I still want to remind these authors to re-calculate their results by reasonably including important details like very recent advances, Searching for novel materials via 4f chemistry (Journal of Rare Earths Volume: 37 Issue: 1 Pages: 1-10 Published: JAN 2019); Hybridized valence electrons of 4f(0-14)5d(0-1)6s(2): the chemical bonding nature of rare earth elements (JOURNAL OF RARE EARTHS Volume 35(8): 837-843, 2017).

Reviewer #2 (Remarks to the Author):

The authors answered the reviewers' questions point by point and revised the manuscript carefully. However, now I still have some questions.

1. Since the FTO substrate are compose of many particles, whose surface areas are large, the value of the capacitance of the substrate should be measured and subtracted.
2. Based on the SEM images, it could be observed that the deposited CeO₂ covers the particles of the FTO substrate and the gaps between the particles are still there. Moreover, the average size of the particles is about 50 nm, which is close to the thickness of the deposited CeO₂ "film". How can the authors obtain the thickness of the deposited "film" with AFM?
3. The Supplementary Table 5 is wrong. The header is "oxidation of Ce³⁺", however, the chemical equation describes the change of Ce⁴⁺ to Ce³⁺. Please correct it and recalculate the energy.
4. In Methods part of "synthesis of ultrathin film", the pH of the solution should be listed.

Reviewer 1

General Comment: I read through all files presented by this work, much better than before.

Response: The authors thank the reviewer for his/her general comment. The response to the specific comment is given below.

Comment 1: I still have one more concern about their "First-Principles Calculations", something fundamental may be clarified if these authors can make this try.

For example, the electrons as valence for Ce should be 4f, 5d, 6s, 6p. Even the bonding conditions for rare earth like Ce are complicated, I still want to remind these authors to re-calculate their results by reasonably including important details like very recent advances, Searching for novel materials via 4f chemistry (Journal of Rare Earths Volume: 37 Issue: 1 Pages: 1-10 Published: JAN 2019); Hybridized valence electrons of 4f(0-14)5d(0-1)6s(2): the chemical bonding nature of rare earth elements (JOURNAL OF RARE EARTHS Volume 35(8): 837-843, 2017).

Response:

Thanks for very helpful comment because it drew our attention to an error in the technical description of our first-principles calculations based on density functional theory (DFT). Specifically, in our DFT calculations, the Ce valence electrons were assigned initially to be 5s, 5p, 6s, 5d, and 4f (12 in total) and the O valence electrons were assigned initially to be 2s and 2p (6 in total). However, we mistakenly omitted to include the 5s, 5p, and 5d electrons from the text description although these had been included in the calculations. Consequently, the DFT calculations did in fact consider the effects stemming from the highly localised d and f electronic orbitals. Further, we added the missing text. This addition makes the work consistent with the other papers including the two papers that have been mentioned, which now are cited in the text.

Textual Modification (Page 8 and 13, Main draft, First-Principle Calculations)

Page 8: The mechanism of electron conduction has been investigated using spin-polarised density functional theory (DFT) calculations (for details, see methods). In accordance with recent work^{1,2} and

DFT calculations³, the Ce electrons in orbitals 5s, 5p, 6s, 5d, and 4f and the O electrons in orbitals 2s and 2p were considered to be the valence electrons.

Page 13: The “projector augmented wave” method was used to represent the ionic cores by considering the electrons to be valence: Ce: 5s, 5p, 6s, 5d, and 4f; O: 2s, 2p.

Reviewer 2

General Comment: The authors answered the reviewers' questions point by point and revised the manuscript carefully. However, now I still have some questions.

Response: The authors thank the reviewer for his/her positive comments. The responses to each of specific comments are given below:

Comment 1: Since the FTO substrate are compose of many particles, whose surface areas are large, the value of the capacitance of the substrate should be measured and subtracted.

Response:

FTO generally would not be examined for capacitance under the given potential used in the present work. However, to put it to the test, as requested, the electrochemical capacitances of the bare FTO substrate were measured using cyclic voltammetry at three scan rates 20, 50, and 100 mVs⁻¹, which are shown below in Figure 1(a-c), respectively. The potential window used for capacitance measurement is the same as that used for the CeO_{2-x} ultrathin films in the present work (-1 to -0.2 V vs. Ag/AgCl). The results for the volumetric charge of the FTO substrate were observed to be in the range 3 x 10⁻⁹ to 5 x 10⁻⁹ mC, as given in Table 1. In contrast, the cyclic voltammograms of a CeO_{2-x} film, which was deposited at a scan rate 3000 mVs⁻¹ for 100 cycles and measured at scan rates of 20, 50, and 100 mVs⁻¹, are shown in Figure 1(d-f), respectively. The volumetric charges for CeO_{2-x} were measured to be between 0.2 and 0.7 mC, which are nearly eight orders of magnitude (10⁸) greater than those obtained for the FTO substrates, regardless of the critical role of thickness (Table 1). For comparison, the voltammograms of both FTO and CeO_{2-x} films are combined, as shown in Figure 1(g-

i), where the significant differences in peak areas are clear. Further, using the thickness of the FTO (~ 700 nm), the volumetric capacitances of the FTO substrate were calculated to be a factor of $\leq 5 \times 10^{10}$ less than that of the ultrathin CeO_{2-x} film (~ 24 nm). These capacitance values for both the FTO and ultrathin CeO_{2-x} films are given in Table 1.

Figure 1. Cyclic voltammograms of (a-c) Bare FTO substrate, (d-f) CeO_{2-x} ultrathin film, (g-i) Overlaid voltammograms of both bare FTO and CeO_{2-x} ultrathin films obtained at scan rates of 20, 50, and 100 mVs^{-1} .

Table 1. Total volumetric charge and capacitance of CeO_{2-x} and FTO and the percentage contribution from the FTO on the total capacitance of CeO_{2-x} .

Scan Rate (mVs^{-1})	Total Volumetric Charge (mC)		Volumetric Capacitance (Fcm^{-3})		Capacitance Contribution of FTO (%)
	FTO Substrate	CeO_{2-x} Ultrathin Film	FTO (~ 700 nm)	CeO_{2-x} (~ 24 nm)	
20	3×10^{-9}	0.2	0.00000084	1585.99	0.00000005
50	4×10^{-9}	0.4	0.00000047	1406.07	0.00000003

100	5×10^{-9}	0.7	0.00000029	1237.35	0.00000002
-----	--------------------	-----	------------	---------	------------

Owing to the determined negligible impact of the FTO on the total capacitance of the CeO_{2-x} , this contribution can be disregarded. However, in order to illustrate the precision of the calculations, a note has been added to the main text as given below.

Textual Modification (Page 14, Electrochemical Measurements section)

In order to confirm that the total capacitances obtained can be attributed to the ultrathin CeO_{2-x} films, the volumetric capacitance of bare FTO was determined to be insignificant ($\leq 5 \times 10^{-8}$ % of that of the CeO_{2-x} capacitance).

Comment 2: Based on the SEM images, it could be observed that the deposited CeO_2 covers the particles of the FTO substrate and the gaps between the particles are still there. Moreover, the average size of the particles is about 50 nm, which is close to the thickness of the deposited CeO_2 "film". How can the authors obtain the thickness of the deposited "film" with AFM?

Response:

The authors thank the reviewer for this perceptive comment.

The original text provided data for the film thicknesses over only the limited area of TEM images. Consequently, in order to confirm the accuracy and precision of these data, AFM imaging was done over a large area of $\sim 210 \mu\text{m}^2$ at the stepped interfaces between FTO substrates (uncoated darker area on the right side of the AFM image in Figure 2) and the ultrathin CeO_{2-x} films (coated brighter area on the left side of the AFM image in Figure 2). The reliability of the data was increased further by effectively determining areal scan using 128 scan lines to generate the mean profiles rather than the single linear-step technique. The consistency of the thicknesses of the ultrathin CeO_{2-x} films is illustrated using representative raw AFM data accompanied by the corresponding height profile, as shown in Figure 2. To ensure the accuracy of the AFM measurement, the calibration of the AFM scanner was checked using a standard 180nm calibration grid and the height sensor measurement was precise and within 1% variation. The mean thicknesses of the films were evaluated by measuring the overall height difference between the area covered by the thin film and the FTO substrate.

Figure 2. Screenshot image of AFM image and corresponding height profiles for CeO_{2-x} ultrathin films deposited on FTO substrate at scan rate of 300 mVs⁻¹ for 100 cycles.

Further, TOF-SIMS imaging (Supplementary Fig. 9 and 10), which is based on elemental mapping over a large area of 900 µm², confirmed the reliability of the AFM thickness measurements. Going one step further, the optical reflectance method of ellipsometry also was done over the very large area of 2.0 mm², which indicated the thickness homogeneity for individual films. All of the new data are given in Table 2.

Table 2. Thicknesses of CeO_{2-x} ultrathin films obtained by the three methods of ellipsometry, TOF-SIMS, and AFM; thicknesses of FTO substrates obtained by ellipsometry.

No.	Scan rate	Cycle number	FTO thickness (nm)	CeO _{2-x} thickness (nm)		
			Ellipsometry (2000 µm ²)	Ellipsometry (2000 µm ²)	TOF-SIMS (900 µm ²)	AFM (210 µm ²)
1	3000	100	727.6	24	24	21
2	1000	100	756.0	38	33	35
3	300	100	721.3	51	50	50
4	50	100	680.8	67	70	67
5	3000	50	735.7	12	11	16
6	3000	100	727.6	24	24	21
7	3000	200	727.7	31	30	30
8	3000	500	714.0	54	52	51

Since the ellipsometry data provide a fourth confirmatory set of thickness measurements, these data have not been included to the main drafts. Nonetheless, details of the process are given as follows:

The measurement was done using a Nanocalc-2000-UV/VIS/NIR (Ocean Optics, FL, USA) instrument using deuterium light with wavelengths in the range 250-450 nm. The light-covered area of the sample surfaces was 2.0 mm². In order to simulate the experimental spectrum, *NanoCalc* software was used. Prior to measurement, the instrument was calibrated using a bare FTO substrate as the reference sample. The thicknesses of both the functional layer (CeO_{2-x}) and the sublayer (FTO) were calculated by taking the measurements in reflection mode at three random points on the surface of each sample.

Comment 3: The Supplementary Table 5 is wrong. The header is "oxidation of Ce³⁺", however, the chemical equation describes the change of Ce⁴⁺ to Ce³⁺. Please correct it and recalculate the energy.

Response:

The authors thank the reviewer for noting the mistake in the Table heading. As for the calculation, Supplementary Table 5 shows thermodynamic calculations and results for the energy required for the (Ox₁-Re₁) redox reaction, leading to the relevant potential (E_{1/2}). This redox reaction includes both Ox₁, which is attributed to the oxidation of Ce³⁺ to Ce⁴⁺, and Re₁, which is attributed to the reduction of Ce⁴⁺ to Ce³⁺. Thus, the theoretical potential (standard cell potential) can be obtained according to the Nernst equation to generate the theoretical E_{1/2} by calculation of the total Gibbs free energy using either the reduction or oxidation reaction. Experimentally, the standard typical method to determine the required energy and the corresponding potential is done by averaging the peak potentials of the redox reaction (Ox₁ and Re₁), hence $E_{1/2} = \left(\frac{E_{Ox} + E_{Re}}{2}\right)$; this is given in Supplementary Table 5. Consequently, the calculations are sound but we have clarified the issue by correcting headers of Supplementary Tables 5, 6, and 7.

Comment 4: In Methods part of "synthesis of ultrathin film", the pH of the solution should be listed.

Response:

As requested, the pH value was included in the "synthesis of ultrathin film" section.

References:

1. Xue, D., Sun, C., Chen, X. Hybridized valence electrons of $4f^{0-14}5d^{0-1}6s^2$: the chemical bonding nature of rare earth elements. *J. Rare. Earths.* **35**, 837-843 (2017).
2. Sun, C., Li, K., Xue, D. Searching for novel materials via 4f chemistry. *J. Rare. Earths.* **37**, 1-10 (2019).
3. Albuquerque, A.R., Bruix, A., Sambrano, J.R., Illas, F. Theoretical study of the stoichiometric and reduced Ce-doped TiO₂ anatase (001) surfaces. *J. Phys. Chem. C.* **119**, 4805-4816 (2015).

REVIEWERS' COMMENTS:

Reviewer #1 (Remarks to the Author):

It is suitable for acceptance for publication without further changes.

Reviewer #2 (Remarks to the Author):

The authors almost answered the questions. Now I have two suggestions:

1. Please describe how they carried out the AFM experiments. In the current manuscript, there are only the AFM results without any descriptions about the experimental processes.

2. I do not think ellipsometry is a suitable technique for measuring the thickness of the films like this. In fact, ellipsometry is an indirect technique to obtain the thickness of a film because the result is "calculated" instead of "measured". The calculated results strongly depend on the optical model which is employed to fit the ellipsometric data and on the optical constants (n and k) of the materials which are very difficult to obtain. Therefore, I do not think the ellipsometric results are reliable here. If the authors insist on providing the results of ellipsometric measurements, please describe the optical model they used and how they obtain the optical constants of the materials.

Reviewer 1

General Comment: It is suitable for acceptance for publication without further changes.

Response: The authors thank the reviewer for his/her recommendation for publication in *Nature Communications*.

Reviewer 2

General Comment: The authors almost answered the questions. Now I have two suggestions.

Response: The authors thank the reviewer for his/her comment. The responses to the suggestions are given below:

Comment 1: Please describe how they carried out the AFM experiments. In the current manuscript, there are only the AFM results without any descriptions about the experimental processes.

Response: As requested, a detailed description of the AFM measurement method is provided in the main draft (characterisation section on page 13). The added text (Page 13, main draft, Characterization):

The thickness measurements were assessed by atomic force microscopy (AFM; Bruker Dimension Icon SPM, PeakForce Tapping mode). A ScanAsyst-Air probe (Bruker AFM probes) was installed in the AFM holder and used for all measurements. The samples were restrained on the stage using a slight vacuum. The scan area was set at 30 μm x 7.5 μm with an aspect ratio of 4; the pixel resolution was 512 samples/line [slow scan axis] and 128 line [fast scan axis], respectively, for the two dimensions. A slow scan rate of 0.195 Hz was used to ensure accuracy. The peak force was minimized to avoid sample deformation and the feedback gain settings was optimized accordingly. The thicknesses of the thin films were determined using the box option consisting of 128 lines, instead of using a single line.

Comment 2: I do not think ellipsometry is a suitable technique for measuring the thickness of the films like this. In fact, ellipsometry is an indirect technique to obtain the thickness of a film because the result is “calculated” instead of “measured”. The calculated results strongly depend on the optical model which is employed to fit the ellipsometric data and on the optical constants (n and k) of the

materials which are very difficult to obtain. Therefore, I do not think the ellipsometric results are reliable here.

If the authors insist on providing the results of ellipsometric measurements, please describe the optical model they used and how they obtain the optical constants of the materials.

Response: The ellipsometry data are not in the text. These were provided only as confirmation for the three measurement methods that were used to determine the thicknesses. The ellipsometry data were only for the private examination of the reviewer.